# A GABAergic and peptidergic sleep neuron as a locomotion stop neuron with compartmentalized Ca2+ dynamics

Wagner Steuer Costa[1,2,10], Petrus Van der Auwera[1,2,3,10], Caspar Glock[1,2,7], Jana F. Liewald [1,2], Maximilian Bach[1,2], Christina Schüler[1,2], Sebastian Wabnig[1,2,8], Alexandra Oranth[1,2], Florentin Masurat[4], Henrik Bringmann [4,5], Liliane Schoofs [3], Ernst H.K. Stelzer [1,6], Sabine C. Fischer[1,6,9] & Alexander Gottschalk [1,2]

Animals must slow or halt locomotion to integrate sensory inputs or to change direction. In *Caenorhabditis elegans*, the GABAergic and peptidergic neuron RIS mediates developmentally timed quiescence. Here, we show RIS functions additionally as a locomotion stop neuron. RIS optogenetic stimulation caused acute and persistent inhibition of locomotion and pharyngeal pumping, phenotypes requiring FLP-11 neuropeptides and GABA. RIS photoactivation allows the animal to maintain its body posture by sustaining muscle tone, yet inactivating motor neuron oscillatory activity. During locomotion, RIS axonal Ca2+ signals revealed functional compartmentalization: Activity in the nerve ring process correlated with locomotion stop, while activity in a branch correlated with induced reversals. GABA was required to induce, and FLP-11 neuropeptides were required to sustain locomotion stop. RIS attenuates neuronal activity and inhibits movement, possibly enabling sensory integration and decision making, and exemplifies dual use of one cell across development in a compact nervous system.

[1] Buchmann Institute for Molecular Life Sciences (BMLS), Goethe University, Max-von-Laue-Strasse 15, 60438 Frankfurt, Germany. [2] Institute for Biophysical Chemistry, Goethe University, Max-von-Laue-Strasse 9, 60438 Frankfurt, Germany. [3] Functional Genomics and Proteomics Group, Department of Biology, KU Leuven, Naamsestraat 59 - box 2465, 3000 Leuven, Belgium. [4] Max Planck Institute for Biophysical Chemistry, Am Fassberg 11, 37077 Göttingen, Germany. [5] Department of Biology, University of Marburg, Karl-von-Frisch-Strasse 8, 35043 Marburg, Germany. [6] Institute of Cell Biology and Neuroscience, Goethe University, Max-von-Laue-Strasse 13, 60439 Frankfurt, Germany. [7] Present address: Max-Planck-Institute for Brain Research, Max-von-Laue-Strasse 4, 60438 Frankfurt, Germany. [8] Present address: od green GmbH, Passauerstrasse 34, 4780 Schärding am Inn, Austria. [9] Present address: Center for Computational and Theoretical Biology (CCTB), University of Würzburg, Campus Hubland Nord 32, 97074 Würzburg, Germany. [10] These authors contributed equally: Wagner Steuer Costa, Petrus Van der Auwera. Correspondence and requests for materials should be addressed to A.G. (email: a.gottschalk@em.uni-frankfurt.de)

Animals actively stop locomotion in order to await certain events or to avoid potentially dangerous situations. In order to quickly resume locomotion after the stop, they must keep their muscle tone. This is in contrast to phases of behavioral quiescence, or sleep, where vertebrates typically lose their muscle tone and assume a relaxed body posture[1,2]. In limbed animals, multi-layered neuronal systems control locomotion[3]. Central pattern generators (CPGs) in the spinal cord mediate (1) rhythm generation, usually by networks of excitatory neurons that oscillate and cause mutual inhibition via interneurons, and (2) pattern generation that regulates motor neurons (MNs), and thus muscle action, to orchestrate coordinated movements underlying locomotion. For left–right coordination during walking, inhibitory and excitatory commissural interneurons are required[4,5]. Locomotion is triggered by excitatory signals descending from supraspinal regions of the mid-brain or hindbrain (in mammals[6]) or of the brain stem (in tadpoles[7]), which "call" the spinal CPG networks into action. Recently, a class of interneurons in the murine brainstem was shown to induce a stop command for the pattern generation systems[8]. These V2a "stop" neurons project to excitatory and inhibitory spinal cord neurons, inducing locomotion halt likely via inhibition of rhythm-generating neurons. The stop neurons do not reduce muscle tone and do not inhibit MNs. Thus the animal does not collapse but rather can quickly resume locomotion. Equivalents of these "stop neurons" and systems for slowing were identified in non-limbed vertebrates[9,10] and recently also in Drosophila[11], where descending interneurons induce locomotion stop during navigation of odorant gradients, while activity of other neurons causes slowing[12]. However, molecular identities of stop cells are only partly known, and also different organisms appear to use different mechanisms and partly redundant circuitry to induce locomotion stop[13]. Thus it is unclear whether "stop" systems evolved several times or whether a primordial locomotion stop system diversified into the different systems present today in different organisms.

In the nematode Caenorhabditis elegans with its much smaller number of neurons, rhythm generation resides in the MNs. Ventral cord excitatory MNs coordinate the undulatory behavior for forward and backward locomotion (B- and A-class, respectively). They exhibit oscillatory activity patterns that are entrained by proprioceptive feedback as well as bi-directional coupling by premotor interneurons (PINs) in the ventral nerve cord[14–17]. In addition, the AS-class of asymmetric MNs exhibits oscillatory activity, interacts with PINs, and contributes to propagation of the body wave[18].

Activity of vertebrate stop neurons contrasts descending pathways that are active during sleep, which halt locomotion and affect muscle relaxation through inhibitory reticulospinal neurons[19]. Sleep also occurs in C. elegans: (1) Lethargus, also called developmentally timed sleep (DTS), occurring during larval molt transitions[20], and (2) stress-induced sleep (SIS), in response to cellular insults[21,22]. Both states are similar with a lack of locomotion and feeding, as well as increased arousal threshold. They are regulated by distinct neuropeptidergic pathways and neurons, i.e., NLP-8, FLP-24, FLP-13 (neuropeptide-like protein, FMRFamide-like peptide) neuropeptides and the ALA (anterior lateral A) neuron for SIS[20,23], while the GABAergic and neuropeptidergic RIS (ring interneuron S) neuron, as well as NLP-22 neuropeptides, are required for DTS[24,25]. RIS photoactivation induced lethargus in larvae, independent of GABA, while FLP-11 neuropeptides were required for RIS-induced sleep, and their overexpression sufficed to cause lethargus[26]. In Ascaris, FLP-11 was shown to be inhibitory to muscle cells[27]. Also in other species, sleep and arousal are encoded by neuropeptidergic and biogenic amine neurotransmitters[28]. In mammals, sleep is mainly

regulated in the ventrolateral preoptic nucleus, where GABAergic and galanin-releasing neurons inhibit orexin/hypocretin-releasing neurons during sleep, while these modulators are released during (and promote) wakefulness[29]. Mutations in the orexin receptor or abnormal activity in the hypothalamic area may lead to narcolepsy or catalepsy[30]. Orexin homologs were found in insects (allatotropin) but not in C. elegans, where instead FLP-21 and NLP-49 neuropeptides as well as pigment-dispersing factor mediate arousal[31–34]. Stop neurons were not identified in C. elegans to date.

Adult C. elegans show behavioral quiescence during satiety, starvation, or recovery from stress, e.g., heat shock[22,23,35]. These states are associated with the RIS and ALA neurons, but it is unclear whether RIS has functions other than as a pure sleep neuron. In the compact C. elegans nervous system, neurons often multitask, thus RIS might be utilized for additional functions. C. elegans frequently interrupts its predominantly forward locomotion by brief reversals, then resumes forward locomotion with a change in direction. It is only partially understood which neurons orchestrate this pirouette behavior and in which sequence they may act[36,37]. Forward locomotion slows down before the animal briefly stops, and this part of the pirouette could be actively controlled by neuronal activity[38]. Thus, may the RIS neuron function like vertebrate stop neurons, i.e., inducing a brief locomotion stop while maintaining muscle tone, to enable directional changes? Furthermore, may functions of sleep and stop neurons have been combined in one cell in the compact worm nervous system, and could this thus represent an evolutionary ancient mechanism from which sleep and stop systems diversified into distinct systems?

We address these questions by (opto-)genetics and by imaging activity of RIS during locomotion. Genetic ablation of RIS reduces reversal and stop events. Specific RIS photoactivation induces a full locomotion stop and also affects other rhythmic behaviors, like pharyngeal pumping. RIS stimulation is accompanied by sustained $Ca^{2+}$ levels in body wall muscle (BWM) cells and inhibition of oscillatory activity in MNs. RIS-dependent behavioral responses are largely blocked without neuropeptidergic transmission, while interfering with GABA transmission affects their kinetics, and eliminating gap junctions uncovers further functions of RIS within circuits coordinating forward/backward transitions. The major determinant of RIS effects in adults, just as in larval sleep, is the FLP-11 neuropeptide. RIS shows compartmentalized axonal $Ca^{2+}$ transients. In the nerve ring process, the onset of these signals correlates with the onset of locomotion slowing, while in an axonal branch, they are correlated with the induction of reversals and require FLP-11 signaling. Our work dissects the function of a C. elegans "stop" neuron, providing new insights into the roles and circuits of such neurons. It may help to understand such neurons, identified only phenotypically[12], and emphasizes that stop cells may exist widely across locomotion systems.

## Results

### Single-cell-specific expression and photoactivation of channelrhodopsin-2 (ChR2) in RIS induces locomotion stop.
We achieved conditional expression of ChR2 and green fluorescent protein (GFP) in the single RIS neuron. Animals showed fluorescence in a single cell body located in the ventral ganglion, next to the pharyngeal posterior bulb, on the right side of the head. RIS has a single process extending anteriorly toward the nerve ring, with a short branch reaching into the ventral nerve cord, while the axonal process wraps around the isthmus of the pharynx (Fig. 1a). When animals expressing RIS::ChR2 were cultivated in presence of all-trans retinal (ATR) and illuminated with blue light, all locomotion behavior stopped: On average, velocity dropped not only by 80% within 4–5 s ($\tau = 1.67$ s; Fig. 1b;

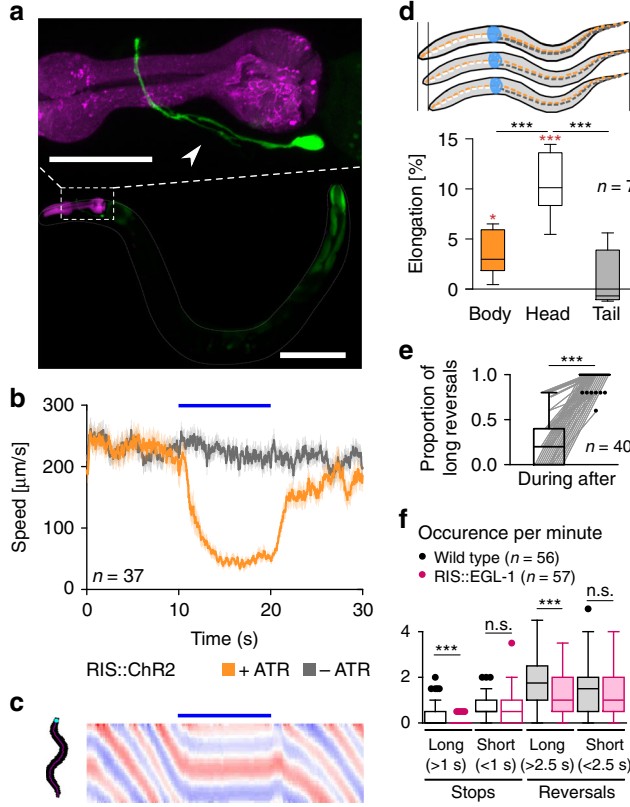

**Fig. 1** Photo-depolarization of the RIS neuron inhibits locomotion.
**a** Maximum intensity projection showing single-cell GFP expression in RIS. The pharynx expressing mCherry is shown in magenta. Scale bar = 100 μm. Inset: Enlarged head region. Arrowhead: RIS axonal branch region. Scale bar = 25 μm. **b** Mean locomotion speed before, during, and after RIS::ChR2 photoactivation (blue bar). Data: mean ± SEM; n animals, cultivated with or without ATR, as indicated. **c** Kymographic representation of bending angles along the spine of a single animal (top: head, bottom: tail, blue to red scale encodes the ventral to dorsal bending). Scale bar is 5 s, blue bar indicates illumination. **d** Analysis of anterior or posterior body elongation during RIS photoactivation, demarcated by a dot painted on the body of the animal (pictogram: blue paint; dotted lines represent entire body (orange), head (white), or tail (gray) lengths along body mid line) in comparison to whole-body analysis. Boxplot with Tukey whiskers; comparisons are to the no light condition (red asterisks) or between body regions (black asterisks). **e** Fraction of reversals larger than one body length after mechanical stimulation to the head region during and after RIS photoactivation. Each animal was tested five times during both conditions (N = 40). **f** Frequency of long and short (shorter or longer than 1 s or 2.5 s, respectively) stops and reversals was compared in wild-type animals, as well as in animals lacking RIS due to expression of the apoptosis-inducer EGL-1. Boxplot with Tukey whiskers. n = number of animals. ***p ≤ 0.001; *p ≤ 0.05; statistical significance tested by one-way ANOVA with Tukey Multiple Comparison test in **d** and Wilcoxon matched pairs test in **e**, as well as by unpaired T test in **f**

Supplementary Movie 1) but also quite immediate for individual animals, i.e., 1–2 s (Supplementary Fig. 1A). We analyzed the body posture of the animals, i.e., bending angles demarcated by three adjacent points along the body axis (Fig. 1c)[39]. Forward locomotion (sinusoidal body wave propagating antero-posteriorly) stopped upon RIS photostimulation: During the 10 s light stimulation, bending angles were "frozen", i.e., the body posture was maintained. This contrasts the stop in locomotion upon photostimulation of all GABAergic MNs[40], where animals

resumed behavior after a few seconds of photostimulation, though being uncoordinated (Supplementary Movie 2). GABA neuron photostimulation causes overall body elongation by 4%[40]. Also RIS photoactivation induced body elongation, however, only ca. 2.5%, affecting only the anterior third, which elongated ca. 10% (Fig. 1d). Thus RIS may inhibit neurons driving the undulations, with some anterior muscle relaxation. RIS photoactivation also attenuated responses to mechanical stimulation, where a harsh touch to the head region led mostly to short reversals smaller than one body length; conversely, after RIS photoactivation, almost all animals reacted with a long reversal (Fig. 1e). During DTS, mechanical stimulation was shown to activate RIS, likely to suppress wake behaviors[41], thus RIS photoactivation might mimic this larval behavior inhibition. To ask whether RIS is sufficient or required for reversals and stops, we ablated it using cell-specific overexpression of the apoptosis inducer EGL-1 ("egg-laying defective"), with GFP as a marker[22]. Animals lacking RIS showed significantly fewer long reversals and stops than wild type (WT; Fig. 1f), demonstrating a requirement of RIS for these behaviors.

Last, rhythmic pumping of the pharynx, the muscular feeding organ, ceased during 30 s RIS photoactivation depending on light intensity (Supplementary Fig. 1B). In electropharyngeograms (EPGs), i.e., extracellular recordings of electrical activity associated with pharyngeal contractions[42], RIS photostimulation generally evoked a complete absence of electrical transients and on average significantly reduced the number of pump events (Supplementary Fig. 1C, D). Thus RIS activation, presumably through GABA and neuropeptide release, may inhibit pharyngeal pumping during locomotion reversals, in parallel to another established pathway for pumping inhibition using serotonin, the SER-2 receptor, and $G\alpha_O$ signaling[37].

**RIS suppresses oscillatory activation of BWM by affecting cholinergic neurons.** To explore the effects of RIS on the locomotion system, we used $Ca^{2+}$ imaging. RCaMP1h, a red-fluorescent genetically encoded $Ca^{2+}$ indicator[43], was specifically expressed in BWMs in addition to ChR2 in RIS, and animals were physically immobilized[44] (Fig. 2a, Supplementary Movie 3). $Ca^{2+}$ signals were monitored over time, either in dorso-ventrally opposite regions (Fig. 2a, b) or in line scans along the animals' dorsal muscles, assessed as "kymograms" spanning the body length (Fig. 2a, c). Despite the absence of dynamic proprioceptive feedback, muscular $Ca^{2+}$ levels visibly fluctuated along the body. $Ca^{2+}$ levels were high in bent regions, in line with muscular activity underlying the body bend, and with the proprioceptive coupling of MNs in one body segment to the anterior segment[45]. The $Ca^{2+}$ signals oscillated at low frequency (ca. 0.12 Hz) in a dorso-ventrally reciprocal fashion (Fig. 2b). They could thus reflect rhythmic activity of the cholinergic MNs innervating the respective muscle cells. Oscillations were much slower than in free-moving animals (~0.36 Hz;[46] note that, due to immobilization, no traveling of the $Ca^{2+}$ wave is observed). During RIS ChR2 activation, this oscillatory activity essentially stopped, while BWM $Ca^{2+}$ levels did not obviously change (Fig. 2c; for Δ [$Ca^{2+}$] over time, see Supplementary Movie 3B). Since oscillations were asynchronous across animals and to enable comparisons independent of actual signal intensities, we used sample autocorrelation, where the autocorrelation period reflects the extent of oscillatory activity (Fig. 2d). For animals grown without ATR, the period duration did not differ between the dark and lit periods, while it significantly increased with ATR present (Fig. 2e), indicating robust slowing of oscillations (since during 20 s stimulation often no full oscillations occurred, we assumed a 20 s minimum of the period).

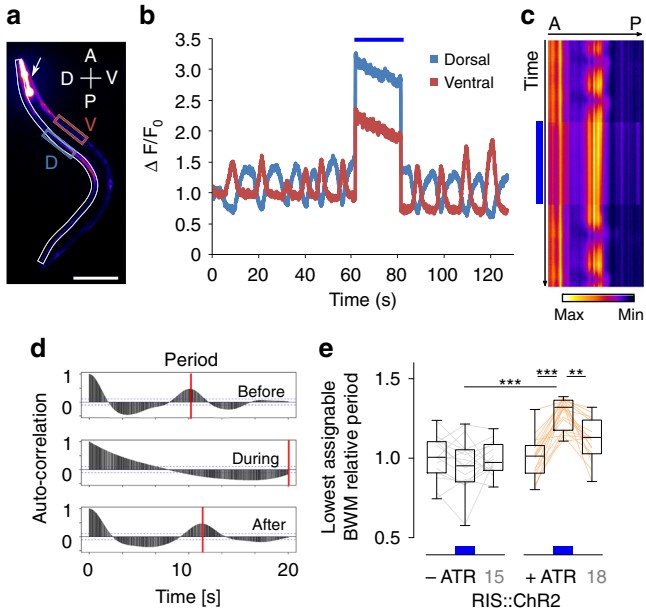

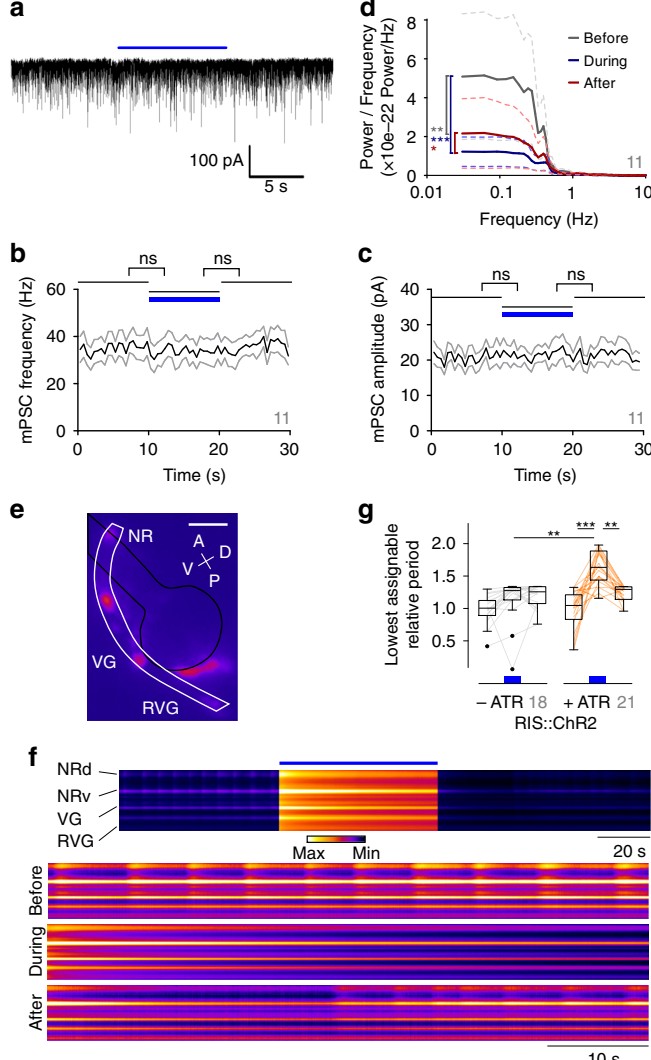

**Fig. 2** RIS photoactivation stopped muscular $Ca^{2+}$-dynamics. **a** Maximum intensity projection of RCaMP imaging in BWM cells of an immobilized animal (arrow: pharynx, p*myo-2*::mCherry marker). Boxed regions: Violet, blue: regions of interest for dorso-ventrally alternating activity; white: region of interest for kymographic analysis of dorsal muscle $Ca^{2+}$ signals along the body in **c**, scale bar = 100 μm, A, P, V, D: anterior, posterior, ventral, dorsal, respectively. **b** $Ca^{2+}$ dynamics in both regions of interest from **a** (V = ventral, D = dorsal muscle cells) measured before, during, and after RIS::ChR2 photostimulation, denoted by the blue bar. $F_O$ defined as the mean RCaMP intensity during the first 4.5 s of the recording. **c** Kymograph representation of the $Ca^{2+}$ dynamics along the dorsal side. The RCaMP signal was normalized for visualization purposes. RIS photoactivation: blue bar. Scale bar = 10 s. **d** Example of the RCaMP signal autocorrelation for a specific point in the BWM over time, before, during, and following illumination. Ventral and dorsal BWMs were analyzed. Red lines: Peak autocorrelation of two consecutive $Ca^{2+}$ waves and their time lag (if no consecutive $Ca^{2+}$ signals were detected during the stimulation period, stimulus duration was taken as lower bound). **e** Distribution of the mean change in the muscular $Ca^{2+}$ oscillation period per animal, compared to before RIS photoactivation. When no oscillations occurred, the duration of photostimulation (20 s) was assumed as minimal period. Compared are animals without and with ATR, number of animals indicated in gray numbers. \*\*\*$p \leq 0.001$; \*\*$p \leq 0.01$; statistical significance tested by ANOVA, Bartlett's test, and Bonferroni's multiple comparison test

Next, we assessed these effects in body muscle, i.e., downstream of MNs, by electrophysiology. Miniature postsynaptic currents (mPSCs) report on release of neurotransmitter from single synaptic vesicles (Fig. 3a). If RIS photodepolarization reduced muscle activity by inhibiting MNs, we would expect reduced mPSC rates. However, RIS::ChR2 photoactivation neither abolished nor reduced mPSCs and their frequency or amplitude (Fig. 3a–c). This indicates that MNs did not reduce their activity during RIS signaling and released the same amount of transmitter. Possibly they altered their relative activities, e.g., if the frequency of oscillation of the CPGs they constitute is altered, and thus MN activity may have become asynchronous. We thus analyzed the frequencies of the observed mPSCs (Fig. 3d). Predominantly low frequencies were populated in this analysis, and RIS photoactivation caused a significant reduction of their power. We also assessed muscular action potentials, which occurred at low frequency, not

**Fig. 3** RIS photoactivation suppressed motor neuron (MN) synchrony and $Ca^{2+}$ oscillations. **a** Exemplary voltage clamp recording of BWM cell, postsynaptic to MNs. Blue bar denotes RIS::ChR2 photostimulation **b**, **c** Analysis (mean ± SEM) of mPSC frequency (**b**) and amplitude (**c**). Blue bar: Illumination period; $n = 11$ animals. **d** Fourier transform with multi-taper analysis of mPSC events across the observed frequencies. Mean (solid lines) ± SEM (dashed lines) of the periods before, during, and after RIS photostimulation of $n = 11$ animals. **e** RCaMP fluorescence in cholinergic neurons in the head with region of interest from dorsal nerve ring (NR; d and v denote dorsal and ventral portions in **f**) through ventral to retrovesicular ganglia (VG, RVG) marked in white; black: pharynx outline. Scale bar = 25 μm. For identity of cells imaged, see Supplementary Fig. 3A. **f** Kymograph representation of cholinergic neuron $Ca^{2+}$ dynamics from the dorsal NR to posterior RVG. Scale bars, upper = 20 s, lower = 10 s; blue bar: illumination period, lower three panels show expanded views. **g** Autocorrelation analysis (as in Fig. 2e); distribution of mean change in $Ca^{2+}$ oscillation period in cholinergic neurons, per animal, relative to before RIS photoactivation. When no oscillations occurred, the duration of photostimulation (60 s) was assumed as minimal period. Number of animals indicated in gray. \*\*\*$p \leq 0.001$; \*\*$p \leq 0.01$; \*$p \leq 0.05$; statistical significance tested by two-way ANOVA in **d** and ANOVA, Bartlett's test, and Bonferroni's multiple comparison test in **g**

obviously silenced by RIS photostimulation (Supplementary Fig. 2). Thus, muscle tone is maintained during RIS activity, while MN activity is desynchronized.

As the observed muscle activity is evoked by MNs even in restrained animals[47], we wanted to analyze MNs directly. If RIS effects on BWM $Ca^{2+}$ fluctuations occur at the MN level, $Ca^{2+}$ fluctuations in MNs may seize during RIS photoactivation. We expressed RCaMP in a large subset (132/160) of cholinergic neurons, encompassing all synaptic cholinergic partners of RIS (VB, DB, AS, RMD, SMD, SDQ, PVC, AVE, including cells in the head ganglia, but excluding SAB; Supplementary Fig. 3A; see "Methods"), and analyzed $Ca^{2+}$ dynamics before and during RIS photoactivation (Fig. 3e, f; Supplementary Movie 4). We observed spontaneous fluctuations in the signals across head MNs of ~0.17 Hz. RIS photoactivation increased the mean relative autocorrelation period 1.6-fold (Fig. 3g), i.e., activity in the head cholinergic nervous system was significantly reduced. No such effect was seen in animals raised without ATR. Thus RIS::ChR2 activity reduced $Ca^{2+}$ oscillations in cholinergic neurons by slowing oscillatory activity in the neuronal network, which is the likely reason for the observed reduction of muscle oscillations.

**RIS photostimulation effects are accelerated by GABA transmission.** Effects of RIS on various behaviors could depend on different types of neurotransmission: RIS is GABAergic,

peptidergic, and makes gap junctions, which may all be driven by ChR2 stimulation. In RIS' control of DTS, GABA played no role, as lethargus still occurred in GABA-defective mutants, and was instead instructed by neuropeptidergic transmission[26]. We tested mutants of these signaling pathways by analyzing the locomotion state as forward, reverse, or stop categories, defined by a threshold of ±45 μm/s on the velocity trajectory. WT animals stopped locomotion during RIS photoactivation (Fig. 4a). After the stimulus ended, a significant proportion the animals resumed locomotion within 2 s, however, inducing reversals, from which they gradually returned to mostly forward locomotion. Mutants lacking the vesicular GABA transporter ("uncoordinated" *unc-47 (e307)* animals) still exhibited the RIS-induced stop response, with delayed onset, and the post-stimulation reversal. RIS induced body elongation that was significantly stronger in *unc-47* mutants than in WT (Fig. 4b). This appears paradoxical, yet, compared to WT, *unc-47* animals are pre-contracted due to absence of GABA at the neuromuscular junction, thus inhibition of cholinergic dynamics by RIS (Fig. 3) likely causes more pronounced body elongation. In sum, GABA is not involved in maintaining RIS-induced locomotion stop phenotypes or the post-stimulation reversal but rather speeds up the slowing and

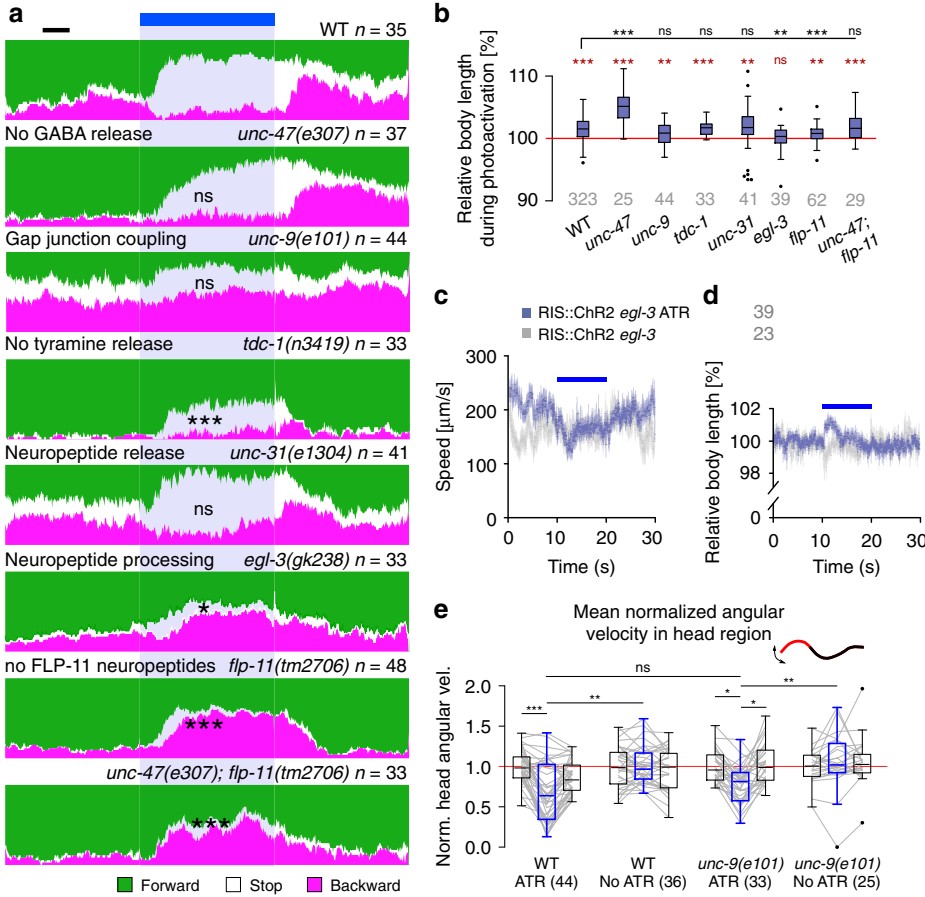

**Fig. 4** The stop phenotype induced by RIS photoactivation requires GABA and neuropeptide signaling. **a** Animal locomotion analyzed before, during, and after photoactivation of RIS (in *lite-1(ce314)* background, to eliminate unspecific photophobic responses) and the proportion of animals in distinct state (forward (green), stop (white), reversal (magenta)) deduced from the animal velocities is represented in color, over time, across all animals analyzed (number of animals and genotypes indicated above each data set). Significant change in stop proportion during RIS photoactivation versus WT indicated; blue bar and blue shade: illumination period; scale bar: 2 s. **b** Relative body elongation during RIS photoactivation; box plot with Tukey whiskers, numbers of animals, and genotypes are indicated below. **c, d** Mean ± SEM locomotion speed (**c**) or body length (**d**) before, during, or after photoactivation of RIS::ChR2 (blue bars) compared in *egl-3(gk238)* mutants raised with or without ATR. Number of animals depicted in gray. **e** Mean normalized angular velocity in anterior quarter of the animal for WT and *unc-9(e101)* mutants expressing RIS::ChR2, with and without ATR. Box plot with Tukey whiskers. ***$p \leq 0.001$; **$p \leq 0.01$; *$p \leq 0.05$; ns: non-significant; statistical significance tested by ANOVA, Kruskal–Wallis with Dunn's Multiple Comparison Test in (**a**; black, versus WT in **b**, **e**) or Wilcoxon Signed Rank Test, versus no body length change (red, in **b**)

stopping effects induced by RIS activation. This may be via RIB neurons, which were shown to increase locomotion speed, and to which RIS makes synapses[48,49] (Supplementary Fig. 3B).

**RIS photostimulation induces subsequent reversals via gap junctions.** RIS forms electrical synapses with five neuron types[49] (Supplementary Fig. 3B): AIB, AVJ, DB, RIM, and SMD. We analyzed whether effects observed during and after RIS::ChR2 activation may be due to concomitant depolarization of these neurons in mutants lacking the innexin (gap junction subunit) UNC-9, expressed in RIS and forming homotypic or heterotypic gap junctions with itself or with UNC-7[50,51]. Based on its expression pattern, lack of UNC-9 should affect gap junctions between RIS and AIB, AVJ, DB, SMD, and RIM[52] (Supplementary Fig. 3C). RIM inhibits reversals[53]. Conversely, reversals can be induced by AIB neurons, which inhibit RIM, via disinhibition[54]. *unc-9(e101)* mutants showed a transient body elongation during RIS photoactivation (Fig. 4b), possibly due to RIS inhibitory transmitter release. However, *unc-9* mutants neither increased stop probability upon RIS photostimulation nor did they induce reversals following RIS stimulation (Fig. 4a). Thus reversals mediated via AIB may be induced through RIS activation and gap junctions to AIB, and inhibitory transmission from AIB to RIM might overcome the putative electrical activation of RIM by RIS. Reversals following RIS stimulation may result from rebound activity upon offset of RIS–RIM electrical stimulation, and a longer-lasting RIS–AIB stimulation, disinhibiting RIM. Our findings for *unc-9* animals are complicated by their low basal locomotion speed. Thus we assessed the angular speed of the head region before, during, and after RIS photoactivation as a proxy for speed (Fig. 4e). RIS activation inhibited head movements in WT and in *unc-9(e101)* mutants. We further analyzed whether RIS may evoke behavior via RIM, which is tyraminergic[53], thus RIS depolarization could affect RIM tyramine release. SMD neurons together with RMD neurons control head movements by activating muscles. RIM/tyramine inhibits RMD, SMD, and head muscles[49,55], thus changing locomotion stop probability[56]. The tyramine-deficient mutant *tdc-1(n3419)* moved almost only forward, and upon RIS photoactivation, animals displayed a significantly reduced propensity to stop (Fig. 3a). Also these animals showed no reversals after the RIS photostimulation period, yet the body elongated (Fig. 3b). In sum, RIS stimulation may affect slowing and subsequent reversals in part via RIM neurons.

**RIS photostimulation effects require neuropeptides.** We next assessed the role of neuropeptides in photoevoked RIS::ChR2 signaling by analyzing mutants lacking the $Ca^{2+}$-dependent activator protein for secretion (CAPS, encoded by *unc-31*) or the pro-protein convertase EGL-3. UNC-31 is required for secretion of (many of the) mature neuropeptides, while EGL-3 mediates processing of most if not all neuropeptide precursors[57]. RIS photostimulation in *unc-31(e1304)* mutants still evoked stopping (Fig. 4a), thus release of neuropeptides mediating RIS effects may require factors other than UNC-31[58]. However, *egl-3 (gk238)* mutants were largely affected and stopped significantly less than WT: Only a very transient (ca. 2 s) and minor speed reduction (~28%) was observed (Fig. 4c), while WT slowed down by ~72% and stopped for the entire 10 s illumination period (Fig. 1b). After a transient elongation during the first 2 s of RIS photoactivation (presumably due to GABA; Fig. 4d), *egl-3(gk238)* mutants on average showed no relaxation (Fig. 4b). RIS was previously shown to regulate DTS using FLP-11 neuropeptides[26]. *flp-11(tm2706)* mutants showed diminished RIS-induced elongation (Fig. 4b) and almost no stops. Instead, *flp-11* animals reversed more, right after RIS stimulus onset (Fig. 4a), as

observed for *egl-3* mutants. FLP-11 peptides may inhibit AVE backward command interneurons via chemical synapses (Supplementary Fig. 3B), and without FLP-11 neuropeptides, AVE may more readily induce reversals, as also the AIB–RIM disinhibitory pathway is activated by RIS. The slowing response was much briefer in *flp-11* mutants, and albeit transient stops were observed, no *flp-11* animal stopped locomotion for the entire stimulus period. RIS activation in *flp-11* mutants also caused no pharyngeal pumping inhibition ($N = 3$, $150 < n < 200$ animals per trial). Since *unc-47* GABA mutants had delayed stop phenotypes, while *flp-11* mutants displayed transient stops at the start of RIS photoactivation, we wondered whether their phenotypes were independent. In *unc-47; flp-11* double mutants, the RIS::ChR2-induced stop was completely abolished; however, reversals still occurred (Fig. 4a). In sum, efficient RIS-mediated locomotion stop is jointly induced by GABAergic and FLP-11 peptidergic signaling. The former modality evokes fast slowing, the latter causes sustained stops.

**Recording of $Ca^{2+}$ activity in the RIS axon and soma in freely moving animals.** Photostimulated RIS affects locomotion, pharyngeal pumping, and withdrawal after mechanical stimuli via GABA, FLP-11, and possibly other neuropeptides acting on different cells. During intrinsic behavior, RIS is active along with other neurons. Singular stimulation of RIS, evoking the observed phenotypes, could thus artificially exaggerate only aspects of behaviors that occur when RIS is (co-)activated with/by other cells. We explored this by analyzing RIS activity in free-moving animals by $Ca^{2+}$ imaging. We modified a previously described tracking system[59]. A four-quadrant photomultiplier controls a stage, keeping a fluorescent spot in the center, and a high-magnification fluorescence video is acquired. An infrared behavior video is acquired at low magnification, stage positions are recorded, and the combined data streams allow behavioral quantification and correlation with neuron activity. We expressed GCaMP6s in RIS, along with a red fluorescent protein in the pharyngeal terminal bulb (TB), to track the head region (Fig. 5a). As not RIS but the TB was tracked, the RIS image rotates around the center of the picture, while the animal changes direction. Custom-written software for image processing (1) registered images on the RIS cell body, (2) cropped a region of interest (ROI) containing the entire RIS morphology and rotated it such that the axon was oriented (Fig. 5b, Supplementary Fig. 5 and Movies 5 and 7), (3) defined a smaller ROI for each image, depicting only the RIS soma and axon, (4) fitted, along the spline of this ROI, a parabola, defined 100 equally spaced perpendicular segments, and quantified their fluorescence.

**RIS axonal $Ca^{2+}$ activity during locomotion is correlated with slowing and the onset of reversals.** RIS $Ca^{2+}$ activity, assessed on short time scales (in contrast to previous analyses during sleep), was transient, or lasted for several seconds, and coincided with slowing and/or reversals (Fig. 5c), for which the magnitude of the $Ca^{2+}$ signal was indicative. $Ca^{2+}$ rose stepwise while the animal slowed, until high levels were reached (Fig. 5d; Supplementary Movie 6) and the animals exhibited subsequent reversals. During those reversals, $Ca^{2+}$ dropped and forward movement resumed a few seconds later. Across all events recorded, the main change in $Ca^{2+}$ signal was observed in the nerve ring region of the RIS axon (Fig. 5e); sometimes activity was also observed in the branch (Fig. 5c, d; Supplementary Movies 7 and 8). Axonal $Ca^{2+}$ transients are expected to be most indicative of relevant neuronal activity, likely correlated with transmitter release. Analyzing axonal events further helped to exclude noise from intestinal fluorescence. Thus we used only the fluorescence of the anterior

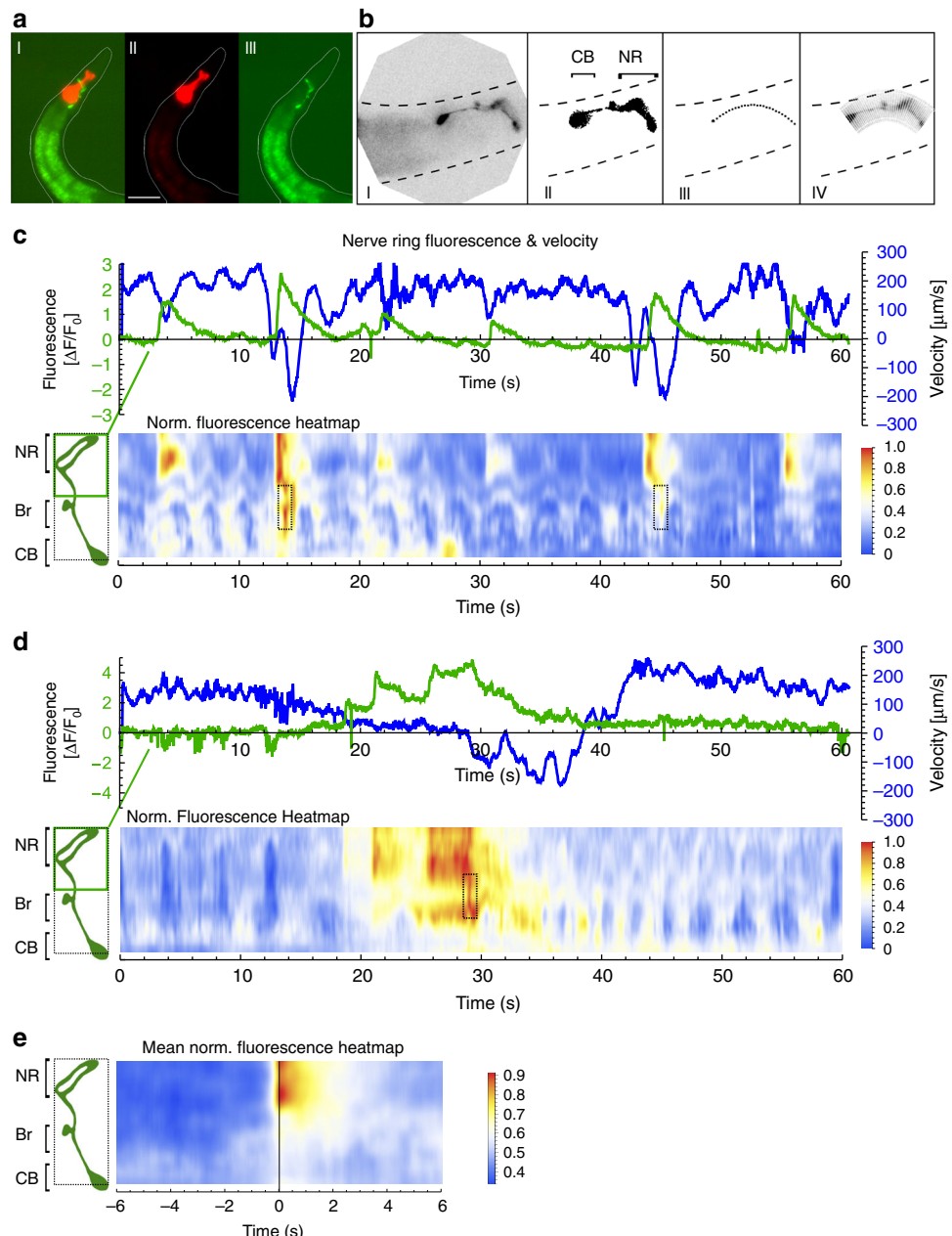

**Fig. 5** Ca²⁺ activity measured along the RIS axon in freely moving animals correlates with slowing and reversals: **a** Strain used for tracking and Ca²⁺ imaging RIS in moving animals, expressing a red fluorescent marker in the pharyngeal terminal bulb (for tracking) and GCaMP6s in RIS. Micrographs of red (II) and green (III) fluorescence and merged color channels (I). Scale bar: 50 μm. **b** Image analysis after binarization and repositioning the soma involved reorienting the raw image (I), masking unspecific gut fluorescence (II), fitting a parabola (III), and measuring fluorescence intensity in perpendicular rectangular ROIs (IV). Dorsal is up, anterior to the left; CB cell body, NR nerve ring. **c**, **d** Upper panels: Representative traces of animal velocity (blue) and fluorescence intensity in the RIS nerve ring portion (green). Lower panels: Corresponding heat maps displaying the normalized fluorescence dynamics along the axon over time. RIS pictograms on the left indicate morphology including nerve ring (NR), branch (Br), and cell body (CB), the distance along the axon as well as the region of the nerve ring (green box) used for calculating the ΔF/F₀ traces in the upper panels, while dashed region shows extent of ROIs analyzed in lower panels (also in **e**). Distinct Ca²⁺ rise events in the branch region are boxed. **e** Mean normalized fluorescence heat map of $n = 45$ acquired Ca²⁺ events along the entire length of RIS, partially excluding the soma, by aligning time windows 6 s prior and post Ca²⁺ peaks ($N = 11$ animals)

half of the neuronal ROI, comprising the axon around the nerve ring, unless we also analyzed the axonal branch. Slowing and reversal events (as shown in Fig. 5c) were identified by analyzing locomotion based on the *x*, *y* position of the tracked animal and its body posture, allowing to derive directional velocity along the mid-body axis. The relative occurrence of reversals and Ca²⁺ peaks are shown in Fig. 6a, as probability distributions, aligned to the nearest Ca²⁺ peak or nearest reversal, respectively. The skew

of these distributions suggested that reversals followed the onset of a Ca²⁺ rise. We aligned events recorded from 20 animals, either to Ca²⁺ peaks (Fig. 6b, d, 45 events) or to the moment of reversal (Supplementary Fig. 5A, 75 reversals). For Ca²⁺ peak-aligned data, we either analyzed velocity (becoming negative upon reversal), to probe if RIS Ca²⁺ may be a determinant of reversals (Fig. 6b), or absolute speed, to explore if RIS may be a speed sensor (Supplementary Fig. 6A).

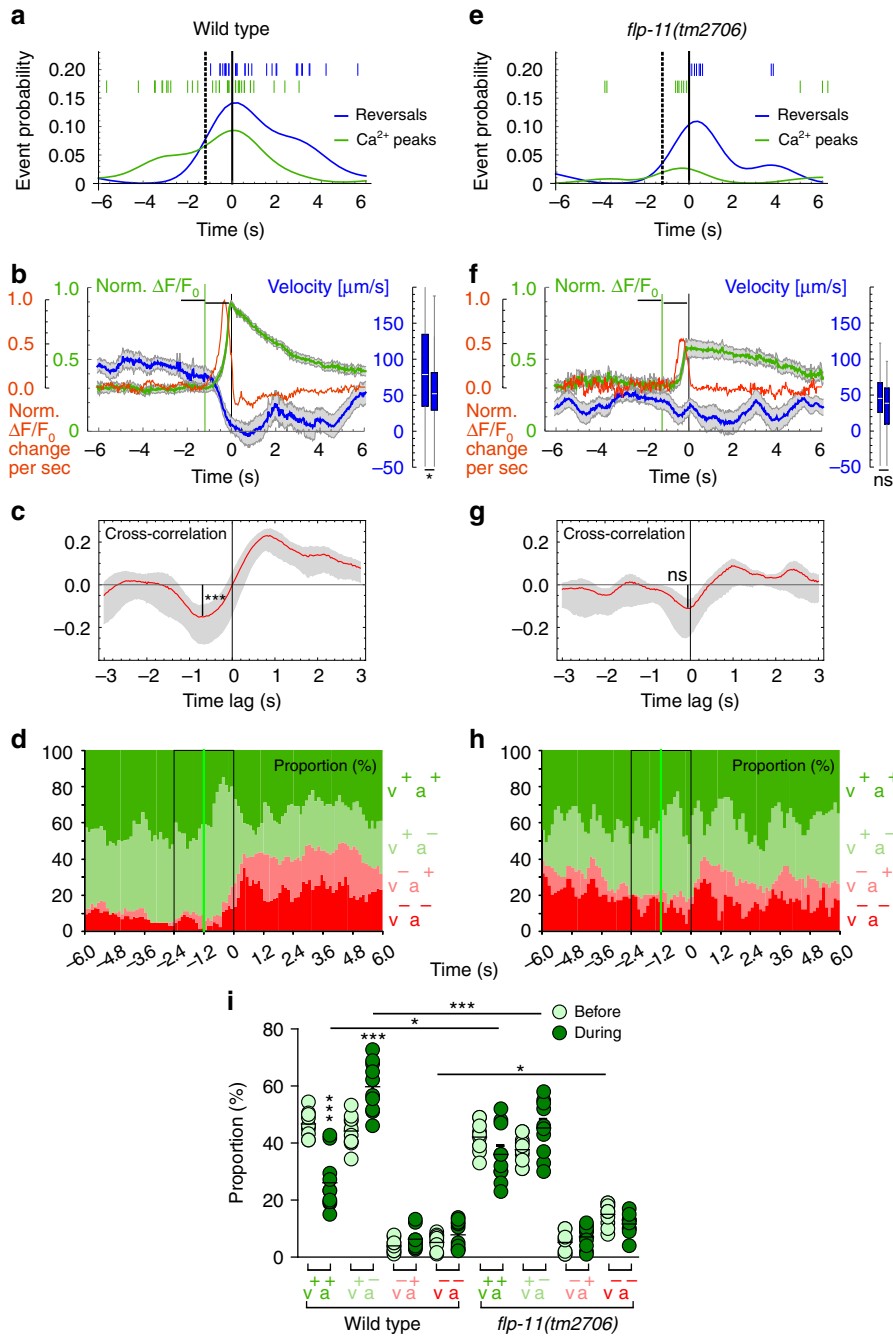

**Fig. 6** RIS $Ca^{2+}$ activity induces decreased forward locomotion and increased reversal probability, which requires FLP-11 neuropeptides: **a** Conditional probability density function of the shortest unbiased time lag to a reversal given a $Ca^{2+}$ peak event (aligned nearest reversal events depicted as blue lines) and vice versa for the probability of a $Ca^{2+}$ peak event given a reversal (green lines: time lag to the nearest reversal) in WT animals. The dotted vertical line indicates the mean onset of a $Ca^{2+}$ rise, the black solid line indicates peak $Ca^{2+}$. **b** $Ca^{2+}$ peak-aligned normalized GCaMP intensity (green, mean ± SEM) in the nerve ring region of the RIS axon (as depicted in Fig. 4c) and animal velocity in μm/s (blue, mean ± SEM, $n = 45$; a significant reduction in the two periods before and during the $Ca^{2+}$ rise is shown on the right, boxplot, $p < 0.001$). Shown in red is the mean first derivative ($dF/dt$ rise rate, $s^{-1}$) of all $Ca^{2+}$ signals. **c** Mean ± confidence intervals of time-shifted cross-correlation (red) of animal velocity aligned to $Ca^{2+}$ signal rise rate, (as in **b**). Anticorrelation is significantly different from 0 and shows a negative time lag ($n = 45$, Pearson's $r = −0.15$). **d** $Ca^{2+}$ peak-aligned analysis of the proportion of the animal population in one of the four behavioral states: 1 moving forward and accelerating ($v^+a^+$, positive velocity and acceleration, dark green; for acceleration data, see Supplementary Fig. 6C, D), 2 moving forward and decelerating ($v^+a^−$, light green), 3 moving backwards, but accelerating ($v^−a^+$, pink), and 4 moving backwards and slowing ($v^−a^−$, dark red). Number of reversing animals increased significantly during RIS $Ca^{2+}$ events ($T$ test, $p < 0.001$). Green line indicates mean onset of RIS $Ca^{2+}$ rise. **i** Scatter plot with means of the data in **d**, **h**; statistical differences analyzed for the time periods indicated by black boxes (before and during $Ca^{2+}$ rise). **e–h** As in **a–d**, but in flp-11(tm2706) background. Boxplots in **b**, **f** compare the average in the time windows indicated by black brackets below the traces. ***$p \leq 0.001$; *$p \leq 0.05$; ns: not significant; statistical significance tested by paired $T$ test in **b**, **f** and unpaired $T$ test in **c**, **g**; ANOVA with Tukey multiple comparisons test in **i**

RIS $Ca^{2+}$ signals required ~1.2 s from detectable onset to peak (Fig. 6b). Concomitantly, a drop in velocity occurred, which in 80% of cases led to a reversal (mean velocity thus approached zero but remained positive). Animals significantly decreased velocity (Fig. 6b, box plot). We also assessed the onset of $Ca^{2+}$ rise (determined from d$F$/d$t$), which likely coincides with transmitter release (Fig. 6b). Cross-correlation of the $Ca^{2+}$ rise rate and velocity drop showed a low but significant coefficient of −0.15 and a time lag of −0.8 s (Fig. 6c). Thus the onset of the $Ca^{2+}$ rise (green vertical line, Fig. 6b) preceded slowing, which is likely regulated by RIS. Correlating the moment of reversal and the $Ca^{2+}$ signal indicated that RIS $Ca^{2+}$ determines duration of reversals (Supplementary Fig. 5A). In sum, a rise in RIS $Ca^{2+}$ preceded slowing and reversals, as also indicated by the event probabilities (Fig. 6a). We conclude that RIS activity precedes the behavioral change and may determine it.

We further analyzed how RIS activity is associated with behavioral changes by assessing the fraction of animals performing particular locomotion behaviors at and around the timing of RIS $Ca^{2+}$ peaks (Fig. 6d, i). Before RIS became active, there was a similar propensity for animals to accelerate or to slow down. During the $Ca^{2+}$ rise, animals were more likely to decelerate, and following the $Ca^{2+}$ peak, reversal probability increased. Thus the likely moment of RIS releasing GABA and FLP-11 neuropeptides is correlated with, and most likely induces, inhibition of locomotion, preceding reversals, analogous to our observations upon RIS photostimulation (Fig. 1b). We conclude that RIS fulfills a crucial role in neuronal programs controlling forward–backward transitions.

**FLP-11 neuropeptides are required for RIS' effects on loco-motion speed.** RIS affects locomotion slowing and is particularly active before reversals. As FLP-11 neuropeptides were required for stopping, we asked whether in *flp-11* mutants RIS $Ca^{2+}$ activity occurs but may be unable to evoke behaviors. From 24 *flp-11* animals, we recorded 100 reversals (Supplementary Fig. 5B) and 25 $Ca^{2+}$ events (Fig. 6e–h). Overall occurrence of reversals or $Ca^{2+}$ events did not differ between WT and *flp-11* (Fisher's exact tests, two sided, $p = 0.865$ and 0.055, respectively, n.s.), thus network functions appeared normal. As in WT, *flp-11* mutants displayed similarly skewed probability distributions for reversals and $Ca^{2+}$ peaks (Fig. 6e). Reversal induction per se is independent of FLP-11. Slowing, on average, was less pronounced during RIS $Ca^{2+}$ rise events, which were less frequently paired to slowing or reversals; thus reversal velocity (Fig. 6f) and speed (Supplementary Fig. 6B) were not significantly different before and during the $Ca^{2+}$ rise. No significant cross-correlation of the $Ca^{2+}$ rise rate and velocity was found (Fig. 6g). *flp-11* mutants did not show significant changes in the proportion of forward, reverse, accelerating, or slowing subsequent to a $Ca^{2+}$ transient, while these proportions were significantly different between WT and *flp-11*, and *flp-11* animals reversed more prior to RIS $Ca^{2+}$ signals (Fig. 6d, h, i). In sum, locomotion, particularly slowing, is abnormally regulated in *flp-11* mutants. RIS' tight control of the timing of reversals after a stop requires FLP-11 release, which also aids in sustaining stops (Fig. 4a), while it elicits fast slowing by GABA. Yet, as reversals occur after RIS photostimulation even without GABA and FLP-11, additional neurons must partake in inducing reversals.

**Compartmentalized $Ca^{2+}$ dynamics in the RIS axon.** In *flp-11* mutants, RIS was on average less efficient in causing slowing. This was intriguing, since we could still observe occasional reversal events. We asked whether there is a specific feature of RIS $Ca^{2+}$ activity that distinguishes such events. We segregated the RIS

$Ca^{2+}$-aligned events into those paired with a reversal and those that merely led to a velocity reduction or stop. We then assessed $Ca^{2+}$ signals spatiotemporally, from soma to nerve ring, encompassing also the branch, focusing on the 2 s centered on the $Ca^{2+}$ event. To capture dynamic changes, $Ca^{2+}$ signals were normalized to the first 150 ms of this time window, and sig-nificantly different $Ca^{2+}$ levels were color coded (Fig. 7).

In the RIS nerve ring region, we observed $Ca^{2+}$ signals for both types of events, i.e., paired (Fig. 5d, e.g., seconds 28–30; Fig. 7a) or unpaired to a reversal (Fig. 5d, e.g., seconds 21–23; Fig. 7b). Interestingly, events paired to a reversal showed significantly increased $Ca^{2+}$ dynamics in the branch region (Fig. 7a; Supplementary Movies 7 and 8A), which were absent when animals only slowed or stopped (Fig. 7b; Supplementary Movies 7 and 8B). Furthermore, reversal–unpaired events showed a significant reduction in branch $Ca^{2+}$ signals ~750 ms preceding the maximal nerve ring $Ca^{2+}$ signal. In these events, no significant activity was observable in the branch, even when the nerve ring region was active (Fig. 7b). We conclude that RIS axonal $Ca^{2+}$ dynamics are compartmentalized and that the branch region has a specific function in induction of, or concomitant with, reversals. RIS axonal $Ca^{2+}$ in *flp-11(tm2706)* mutants had no significant dynamics during reversal (Fig. 7c), while events paired only to a stop still showed nerve ring activity, though no significant reduction in branch activity was observed (Fig. 7d). FLP-11 neuropeptides may directly or indirectly provide positive feedback to RIS during reversals.

## Discussion
Neuronal circuits regulate and fine-tune locomotion. While in mammals this is orchestrated by whole-brain systems like motor and prefrontal cortex, cerebellum, and spinal cord neurons, much fewer neurons must fulfill these tasks in compressed nervous systems. Here we analyzed the role of one neuron, RIS, which orchestrates locomotion slowing and reversals in *C. elegans*. RIS does this by re-employing a peptidergic pathway used in sleep control, e.g., after larval molts or in response to stress. Here RIS uses also GABAergic signaling, not required during DTS induc-tion[26]. During optogenetic stimulation, GABAergic signaling from RIS induces the locomotion stop within seconds, possibly by inhibiting RIB neurons[48] (Supplementary Fig. 3B, C), and sus-tains it by FLP-11 signaling (Fig. 8a). The observed axonal compartmentalization of $Ca^{2+}$ dynamics during locomotion (Fig. 8b) jointly with the requirement of peptidergic and GABAergic signaling suggest that RIS uses both modalities in a spatiotemporally defined manner to control locomotion. RIS' role in both locomotion and sleep control represent two types of temporally different activity: (1) where RIS is employed to control locomotion with brief activity bouts (seconds), in coordination with other locomotion neurons, and (2) where DTS and SIS involve long-term RIS activity (minutes to hours), together with the sleep neuron ALA[22].

RIS depolarization in the nerve ring results in fast GABA- and FLP-11 neuropeptide-mediated inhibition of neuronal activity and locomotion. Suppression of branch depolarization is permissive for locomotion halting, and conversely, if the RIS axon is not hyperpolarized in the branch before depolarization in the nerve ring, the most likely behavioral outcome is a reversal. This coincides with optogenetic experiments where RIS depolarization led to reversals and required both GABA and FLP-11 release. These transmitter(s) dampened cholinergic MN oscillations and desynchronized acetylcholine release, thus suppressing oscillations of body muscle activity. Occasionally reversals occurred also without RIS activity or in the absence of FLP-11. Thus RIS is sufficient but not essential for reversals and

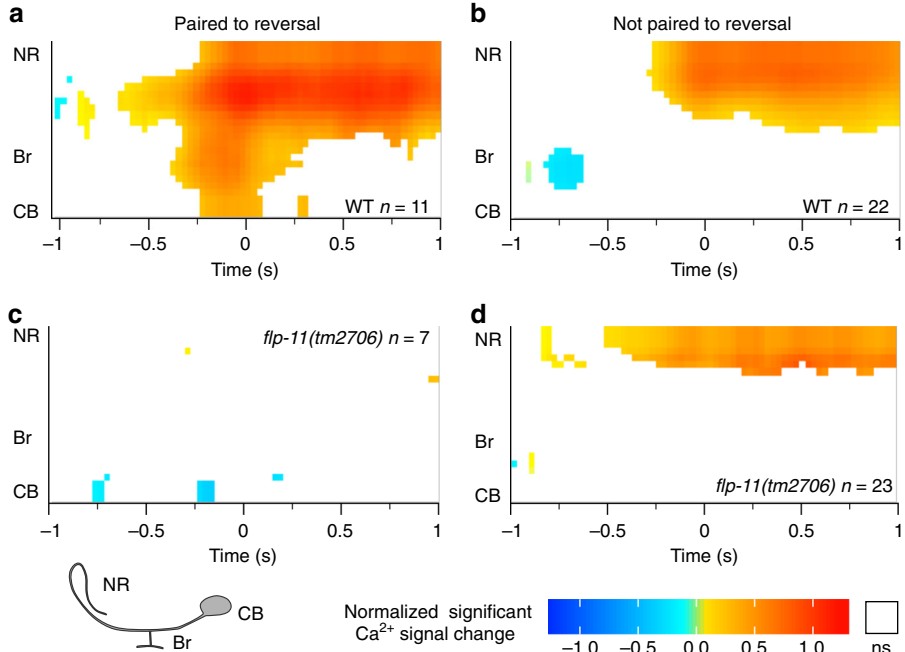

**Fig. 7** Compartmentalized Ca$^{2+}$ dynamics in the RIS process. **a**, **c** Ca$^{2+}$ events paired to a reversal in the 2 s time window flanking a Ca$^{2+}$ signal obtained from freely behaving animals (WT in **a**, *flp-11(tm2706)* in **c**). Top is the dorsal nerve ring (NR) region, passing through the branch region (Br) to the center of the cell body (CB). Only events significantly different from inactivity are shown, color coded for intensity, normalized and relative to the mean of the first 150 ms depicted. Blue and red hues indicate significant reductions and increases, respectively. Non-significant dynamics are omitted and shown in white. **b**, **d** Ca$^{2+}$ events unpaired to a reversal in the 2 s time window flanking a Ca$^{2+}$ signal, as in **a**, **c**. Number of events indicated. Significance level: *T* Test, $p < 0.05$

must act in coordination with other neurons. Roberts et al.[38] described a four-state model of *C. elegans* locomotion, where forward–reverse and vice versa transitions were connected by brief pause states. RIS may induce this brief stop. Hints of this are found in our analyses, e.g., pause events at seconds 13, 44, and 56 in Fig. 5c, accompanied by RIS Ca$^{2+}$ peaks. Also, in analyses of mean Ca$^{2+}$ events, a brief phase of zero acceleration followed the moment of maximal slowing (Supplementary Fig. 6C).

The functional compartmentalization of the RIS axon indicated that the branch is instructive for reversal onset. Since the effects of RIS activation are rather fast and the RIS axon extends only in the nerve ring, the targets of transmitters released by RIS are likely in the head ganglia. RIS synaptic output and input segregate between its nerve ring and branch regions[49], with the branch being mainly postsynaptic, and the nerve ring process being dominated by output synapses. The branch connects to only three neuron types: AVJ, PVC, and SMD. SMD neurons are part of the network encoding the amplitude of Ω-turns (named after the body posture adopted transiently during turning)[60]. No behavior was yet associated with AVJ, while PVC is a PIN inducing forward movement[61]. In the locomotion state model[38], PVC and AVE drive forward and backward locomotion, respectively. PVC innervates RIS and RIS has synaptic output to AVE[49], implying a possible sequence of signaling leading from forward (PVC) to a pause (RIS), e.g. mediated by GABA inhibition of the speed neuron RIB, and transition to a reversal, by gap junctions to AIB and possibly by activating AVE (the latter would imply, however, an excitatory connection, possibly through excitatory GABA receptors (like the EXP-1 channel involved in control of the defecation motor program[62]). However, such speculative models will have to be clarified in animals lacking such receptors cell-specifically. During pure slowing events, branch Ca$^{2+}$ activity was

suppressed prior to RIS activation. PVC activity, which may inhibit RIS (e.g., via muscarinic acetylcholine receptors, as PVC is cholinergic), could alter RIS output such that the evoked behavior is a stop instead of a reversal. Lack of FLP-11 neuropeptides uncoupled reversals from RIS Ca$^{2+}$ activity and was permissive for reversal induction during RIS photoactivation. This suggests that RIS contributes to coordinating reversals by controlling the halt duration and that it uses FLP-11 peptides for this purpose.

The kinetics of RIS::ChR2-elicited behaviors may be due to the different diffusion properties of GABA and FLP-11 neuropeptides (GABA is much smaller than FLP-11 neuropeptides). GABA was required for fast slowing within 1 s, while FLP-11 sustained the locomotion stop. RIS::ChR2 depolarization in WT animals increased reversal probability at the stimulus off-set, while *flp-11* mutants increased their reversal probability already during the photostimulation period. Hence, FLP-11 is required to maintain behavior inhibition during RIS activity. The requirement for GABA and FLP-11 signaling for fast induction of a stop and for its maintenance, respectively, is in line with activities of stop neurons in vertebrates in regulating locomotion inhibition and body posture[8,10]. In the mouse, behavior stop signals are not controlled by a single cell type in striatum[63] but by the activity of stop neurons in the brainstem that project to the spinal cord and depress locomotor rhythm generation[8,64]. Similarly, RIS de-synchronized MN activity. Conservation of not only RIS and of FLP-11 neuropeptides in nematodes[65] but also the neuropeptide VF (NPVF) in fish, which can suppress escape behaviors[66], suggests that RIS acts as a stop neuron in parallel to/upstream of the ventral nerve cord CPG system[17]. A comparison among worm, fly, leech, tadpole, zebrafish, lamprey, and mouse (Fig. 8c) of stop and sleep neurons may suggest that, in compact brains like *C. elegans*, both activities are coalescing in one neuron type, RIS, which effects locomotion stop during

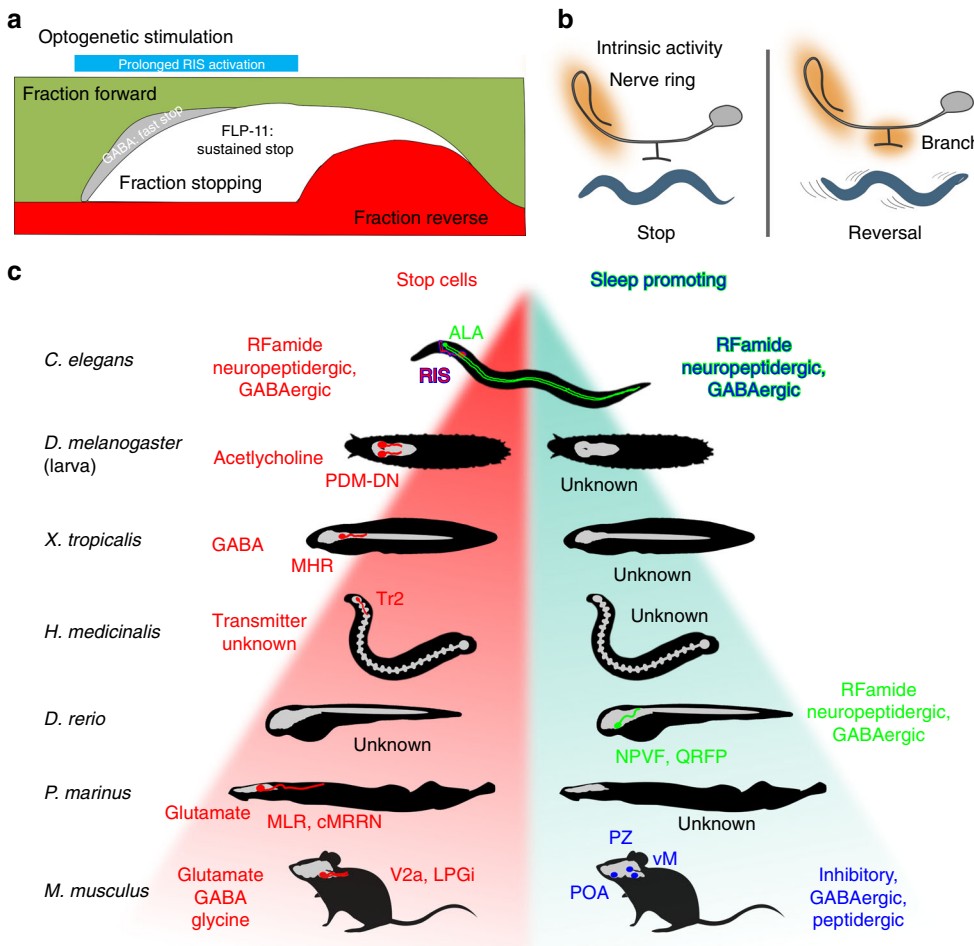

**Fig. 8** Models summarizing stimulated and intrinsic activities of RIS and comparison of sleep and stop neurons across model systems. **a** Fraction of behaviors found in animals before, during, and following optogenetic RIS stimulation (blue bar), categorized as forward (green), stop (gray and white), and reversal (red) movement. Also indicated is the contribution of RIS neurotransmitters to fast and sustained phases of induced stops. **b** Ca[2+] activities of nerve ring and branch regions of the RIS axon, accompanying locomotion behaviors. The nerve ring region is active to induce slowing, while the branch region is additionally active when reversals are induced. **c** Comparison of stop cells[13] and sleep neurons/systems in various model organisms. Hierarchy indicates complexity of the respective brains but implies no phylogenetic relationships. Left: Cells that stop or slow down locomotion when activated, in *C. elegans* (RIS), *D. melanogaster* larvae (PDM-DNs posterior dorso-medial brain lobe descending neurons[11]), *X. tropicalis* tadpoles (GABAergic MHRs, mid-hindbrain reticulospinal neurons), Tr2 cells[70] in the anterior brain of the leech *H. medicinalis*, glutamatergic neurons in the MLR (mesencephalic locomotion region) and RS stop cells of the cMRRN (reticulospinal cells of caudal middle rhombencephalic reticular nucleus) of the lamprey *P. marinus*[10,71], and several types of mammalian stop cells: V2a reticulospinal interneurons in rostral medulla or caudal pons[8], GABAergic neurons in the MLR, GABAergic and glycinergic neurons in the gigantocellular nucleus (GiA), and glycinergic neurons in the lateral paragigantocellular nucleus (LPGi)[72]. No stop cells were identified in zebrafish. Right: Sleep promoting neurons/systems (green: directly, and blue, indirectly promoting sleep, reviewed in ref. [73]) are peptidergic and GABAergic RIS and ALA cells in *C. elegans* (thus combining functions of stop and sleep neurons in RIS); in zebrafish larvae RFamide neuropeptide VF (NPVF), similar to *C. elegans flp-11* derived peptides, inhibits serotonergic neurons in the ventral raphe nucleus; and in the mouse, GABAergic/peptidergic neurons in the preoptic area (POA), inhibitory neurons of the parafacial zone (PZ) of the brain stem, and GABAergic neurons of the ventral medulla (vM) are involved in sleep control. Transmitters used by each cell type are indicated. No sleep systems are known yet for fly larvae, tadpoles, leech, or lamprey. Animal silhouettes were acquired from http://phylopic.org/

brief activity and sleep during prolonged activity. In more complex brains, such functions dissociate into different systems, distinct for sleep control and locomotion stop, which reside in different cell types (molecular identities still need to be determined for most systems) and brain structures, and use different mechanisms. Here more elaborate control regimes have evolved, where sleep neurons induce relaxation, while stop neurons halt locomotion but do not relax muscle. RIS makes this distinction by brief and short signaling via the same molecules. RIS may thus represent a primordial neuronal pathway, e.g., for fast sensory integration, and may have evolved to more finely tuned stop neurons in vertebrates.

## Methods

**Molecular biology**. pCoS6 (pglr-1::flox::ChR2(H134R)::mCherry::SL2::GFP) was a gift from Cornelia Schmitt[67]. The following plasmids were kindly provided by Navin Pokala (Bargmann lab, Rockefeller University, USA): **pNP165** (pSM::pglr-1::flox::ChR2(H134R)::mCherry), **pNP260** (pSM::pnmr1::flox::ChR2::STAR::mCherry), and **pNP259** (pSM::pgpa-14::Cre). **pPD95.79** was a gift from Andrew Fire (Addgene plasmid # 1496; RRID:Addgene_1496). **pOT15** (QUAS::jRCaMP1b), **pOT6** (punc-17::QF), and **pOT7** (punc-4::QS::mCherry) were provided by Oleg Tolstenkov[18]. We used the plasmid **pmyo-3::RCaMP1h** for RCaMP expression in BWM cells[43]. **pmyo-2::GCaMP3** was kindly provided by Karen Erbguth. **pGP** (pmec-4::GCaMP6::SL2::TagRFP) was kindly provided by Douglas S. Kim. **pCG02** (pggr-2::GCaMP6s::SL2::RFP): pGP backbone was ligated to the pggr-2 sequence from pWSC28, both cut with SphI and Not-I-HF. **pCG03** (pggr-2::flox::GCaMP6s::SL2::RFP): pCG02 was linearized with SacII and ligated to the PCR product of pWSC24 with forward primer oCG11 (AAAGAATTCGGTACCGAT

AACTTCGTATAGCATACA) and reverse primer oCG12 (TGGCGCGCCGG GCCCGATAACTTCGTATAATGTATGC). **pWSC05** (pnpr-9): pPD95.79 was cut with SalI and PstI and ligated to a PCR product of pnpr-9 from N2 genomic DNA (forward primer, oWSC09: AGACTGCAGCCGTCAAATGGAAAGGTTCGCGC AT, reverse primer, oWSC10: GTCGACTTCTACGACATTTCCCAGGAAGTAG CTCTAA), cut with PstI and SalI. A nested PCR protocol was run for the promoter fragment with the following outer primers: forward, oWSC13: ACGGAATGT GTCTGCAAAAGAAACGG, reverse, oWSC14: ACATTTCCCAACGACATTT CCCAGG. **pWSC15** (pgrr-1::GFP): pWSC05 was cut with SalI and HindIII and ligated to a PCR product of pgrr-1 from N2 genomic DNA (forward primer, oWSC37: GAAATGAAATAAGCTTAGGCAACCGTGTGCTCTGGC, reverse primer, oWSC38: ATCCTCTAGAGTCGACTCAATAATTAAAGTATGCAG TTGA), cut with HindIII and SalI. A nested PCR protocol was run for the promoter fragment with the following outer primers: forward, oWSC42: AGTGGGGTAAAGCTTGTCCTAGGC, reverse, oWSC43: TCTGCCTGACCCCA GGACGCA. **pWSC17** (pgrr-2::GFP): pWSC05 was cut with SalI and SphI and ligated to a PCR product of pgrr-2 from N2 genomic DNA (forward primer, oWSC40: AATGAAATAAGCTTGCATGCTCTTCCGGCAGATGCGCTGTT, reverse primer, oWSC41: ATCCTCTAGAGTCGACGCCGTCGTGGTAAGAC GTTATAGTT), cut with SphI and SalI. A nested PCR protocol was run for the promoter fragment with the following outer primers: forward, oWSC44: TCTCTCCGCGCTGACCAAGT, reverse, oWSC45: TGGCACCGGTTCGCTCC TACT. **pWSC19** (pgrr-1::Cre): pNP259 was cut with KpnI and HindIII and ligated to pWSC15, cut with HindIII and KpnI. **pWSC20** (pgrr-1::mCherry): pWSC15 was cut with HindIII and SalI and ligated to pNP165, cut with XhoI and HindIII. **pWSC24** (pgrr-2::flox::ChR2(H134R)::mCherry::SL2::GFP): pCoS6 was cut with SalI and SphI and ligated to pWSC17, cut with SphI and SalI. **pWSC28** (pgrr-2:: GCaMP3) pmyo-2::GCaMP3 was cut with MscI and SphI and ligated to the fragment of pWSC17, cut with SphI and MscI.

**Reagents**. Reagents are available upon request.

**PCR for pharyngeal terminal bulb marker**. The red fluorescent mCherry marker in the pharyngeal bulb of the pharynx (driven by the pncx-10 promoter), used for tracking of the region containing RIS, was injected as a linear DNA construct, generated by fusion PCR with these primers: A (GTTCTTTCAACATTGCAAAA AGGCACCA), A' (TACACAGTTGCAGAGGCGTTTAATCAGA), B (ATCTTC TTCACCCTTTGAGACCATTACCTGAAAAAGAAACAGTTGATAAGCG GGT), C (ATGGTCTCAAAGGGTGAAG), D (ACGACGGCCAGTGAATTATC), D' (GGAAACAGTTATGTTTGGTATATTGGG).

**C. elegans cultivation and transgenesis**. *C. elegans* WT (N2, Bristol strain) and transgenic animals were cultivated on either nematode growth medium (NGM) or high growth medium plates seeded with *Escherichia coli* OP-50-1 strain. We noted that, at elevated temperature of 25 °C, conditional transgene expression for RIS was observed also in additional, unwanted neurons, thus we kept animals (including larval stages) at 20 °C at all times. In addition, 100 μM ATR (Sigma-Aldrich) were supplemented to *E. coli* prior to seeding for optogenetic experiments[40]. Transgenic animals were generated following standard protocols. Extra-chromosomal array integration was performed by ultraviolet exposure following standard protocols.

**C. elegans strains**. The following strains were used or generated for this study: N2 (WT isolate, Bristol strain), **HBR1777**: goeIs384[pflp-11::egl-1::SL2-mkate2-flp-11-3'utr, unc-119(+)][22], **ZX1466**: lite-1(ce314)X; zxIs55[pgrr-2::flox::ChR2(H134R):: mCherry::SL2::GFP; pgrr-1::Cre], **ZX1468**: unc-47(e307)III; lite-1(ce314)X; zxIs55 [pgrr-2::flox::ChR2(H134R)::mCherry::SL2::GFP; pgrr-1::Cre], **ZX1469**: unc-31 (n1304)IV; lite-1(ce314)X; zxIs55[pgrr-2::flox::ChR2(H134R)::mCherry::SL2::GFP; pgrr-1::Cre], **ZX1470**: lite-1(ce314)X; zxIs55[pgrr-2::flox::ChR2(H134R)::mCherry:: SL2::GFP; pgrr-1::Cre; pmyo-2::mCherry]; zxIs52[pmyo-3::RCaMP], **ZX1561**: zxIs55 [pgrr-2::flox::ChR2(H134R)::mCherry::SL2::GFP; pgrr-1::Cre], **ZX1577**: lite-1(ce314) X; zxEx360[pgrr-2::flox::ChR2(H134R)::mCherry::SL2::GFP; pgrr-1::Cre], **ZX1891**: egl-3(gk238)X; lite-1(ce314)X; zxEx360[pgrr-2::flox::ChR2(H134R)::mCherry::SL2:: GFP; pgrr-1::Cre], **ZX2017**: zxIs60[pgrr-2::flox::GCaMP6s::SL2::tagRFP; pgrr-1:: nCre]; zxEx378[pncx-10::mCherry], **ZX2099**: flp-11(tm2706)X; lite-1(ce314)X; zxEx360[pgrr-2::flox::ChR2(H134R)::mCherry::SL2::GFP; pgrr-1::Cre], **ZX2140**: unc-47(e307)III; flp-11(tm2706)X; lite-1(ce314)X; zxEx360[pgrr-2::flox::ChR2 (H134R)::mCherry::SL2::GFP; pgrr-1::Cre], **ZX2223**: flp-11(tm2706)X; zxEx1173 [pgrr-2::flox::GCaMP6s::SL2::tagRFP; pgrr-1::nCre; pncx-10::mCherry], **ZX2297**: lite-1(ce314)X; zxEx371[pgrr-1::Cre, pgrr-2::flox::ChR2(H134R)::mCherry::SL2::GFP; QUAS::jRCaMP1b; punc-17::QF; punc-4::QS].

**Optogenetic stimulation and behavioral analysis**. Locomotion parameters of freely moving animals were measured with a single worm tracking and illumination device described earlier[68]. This tracking set-up enabled accurately targeted illumination to specific body segments by a modified liquid crystal display projector, integrated with an inverted epifluorescence microscope. For optogenetics experiments, animals were cultivated in the dark on NGM plates with *E. coli* OP50-1 bacterial culture supplemented with ATR. NGM plates were freshly seeded a day in advance with 250 μl of OP50 bacterial culture supplemented with ATR dissolved in 100% ethanol to a final

concentration of 100 μM. Five minutes prior to analysis, young adult animals were gently picked with an eyelash to unseeded NGM plates under dim red light (>600 nm) and maintained in the dark. Blue light of 470 nm and 1.8 mW/mm² intensity was used to stimulate ChR2 expressed in RIS. Light power quantification was performed with a power meter (PM100, Thorlabs, Newton, NY, USA) at the sample focal plane. All optogenetic experiments followed the same illumination protocol: 10 s tracking and behavioral acquisition (without blue light), subsequently 10 s targeted blue light illumination, followed by another 10 s of tracking without blue illumination. Only the anterior half of the animal was illuminated. Tracks were quality controlled by censoring data points from erroneously evaluated video frames. Briefly, a specialized workflow in KNIME (KNIME Desktop version 3.5, KNIME.com AG, Zurich, Switzerland) allowed data that passed the constraints (animals' speed <1.25 mm/s and length deviation <25%, relative to the first 5 s of acquisition). Animals with >15% of the data points censored were excluded from analysis.

**Anterior and posterior body length analysis**. Single animals were picked onto the center of an unseeded NGM plate and a small spot of ink was painted with an eyelash on the cuticle of the worm, about 1/3 of the length of the animal relative to the head. Worms were recorded during unrestrained locomotion, using a ×10 objective on an Axio Scope A1 (Zeiss). Videos were acquired by a DCC1545M camera (Thorlabs) with 3 s pre-stimulation, 5 s 470 nm at 0.9 mW/mm² RIS photoactivation and 3 s post-stimulation. Analysis was performed in ImageJ, blind to condition, where three frames were manually annotated: one frame before photostimulation, one in the middle of the 5 s photostimulation period, and the last frame of post-stimulation. Two splines were tagged to the center line of the animal, ranging from the head or tail to the ink dot. The length of the dot was also quantified, as well as the length from head to tail. The relative elongation of the head, tail, and full-body splines were calculated in reference to the data before RIS photostimulation.

**Ca²⁺ imaging microscope setup**. An inverted fluorescence microscope (Axiovert 200, Zeiss, Germany) equipped with an MS 2000 motorized stage and PhotoTrack quadrant photomultiplier tube (both Applied Scientific Instrumentation, USA) was utilized for Ca²⁺ imaging, similar to a system described earlier. As excitation light sources, two high-power light-emitting diodes (LEDs; 470 and 590 nm wavelength, KSL 70, Rapp Optoelektronik, Germany) or a 100 W HBO mercury lamp were used. Simultaneous dual-wavelength acquisition was enabled by a Photometrics DualView-Λ beam splitter in combination with a Hamamatsu Orca Flash 4.0 sCMOS camera controlled by HCImage Live (Hamamatsu) or MicroManager (version 1.4.13; http://micro-manager.org) software.

**Ca²⁺ imaging in immobilized worms**. Animals were immobilized on 10% M9 agar pads with polystyrene beads (Polysciences, USA) and imaged either by ×25 (BWM) or ×40 (nerve ring) oil objective lenses. The following light filter settings were used: GFP/mCherry Dualband ET Filterset (F56-019, AHF Analysentechnik, Germany) was combined with 532/18 nm and 625/15 nm emission filters and a 565 long-pass beam splitter (F39-833, F39-624, and F48-567, respectively, all AHF). ChR2 stimulation was performed using 1 mW/mm² blue light. To measure RCaMP fluorescence, 590 nm, 0.6 mW/mm² yellow light was used. The 4 × 4 binned images were acquired at 31 ms exposure time and 20 fps. Light illumination protocols were generated by a Lambda SC Smart shutter controller unit (Sutter Instruments, USA), using its TTL output to drive the LED power supply or to open a shutter when using the HBO lamp. Video synchronization was achieved by cropping the acquisition to obtain equally sized time bins before, during, and after blue light exposure. Image analysis was performed in ImageJ (NIH), called by a custom KNIME workflow, after manual annotation of ROIs. Spline ROIs were selected for the BWM cells or ventral nerve cord to nerve ring region and the kymographs were generated by averaging a $7 \times 7 \times 7$ moving voxel (where time is the third dimension) across the ROI. A ventral and a dorsal BWM ROI were defined, and their periodicity data were averaged for the analysis of a single animal. Additionally, an elliptic ROI was selected for background fluorescence exclusion. Mean intensity values for each video frame were obtained and background fluorescence values were subtracted from the fluorescence values derived for RCaMP. Subtracted data were normalized to $\Delta F/F_0 = (F_i - F_0)/F_0$, where $F_i$ represents the intensity at the given time point and $F_0$ represents the average fluorescence of the first second of the acquisition.

**Signal autocorrelation analysis**. Since spontaneous activity was not synchronized between animals, we introduced the metric of signal autocorrelation for calcium imaging as a means of describing perturbations in the Ca²⁺ dynamics of both BWM and neurons. We observed that Ca²⁺ dynamics in undisturbed animals were periodic, albeit without a fixed frequency. This periodicity could be obtained by analyzing the signal correlation to itself, where peaks in the autocorrelation function were indicative of the period of Ca²⁺ dynamics. This was performed by a custom script in R called by a KNIME workflow. The R script calculated the autocorrelation of a smooth spline fit of the $\Delta F/F_0$ signal and searched for its peaks in window of 20 s of data to calculate the period of the autocorrelation. If no period could be found, it was set as the maximum of 1 in every 20 s. Hence, this function

returned the period lower bound where confidence in the result can be guaranteed. Note that for the autocorrelation analysis only the signal differences over time, and not the absolute values, are taken into account. Thus, this analysis does not require prior signal normalization.

**Axonal $Ca^{2+}$ imaging of RIS in moving animals**. For measuring spontaneous $Ca^{2+}$ activity, a tracking system for single neuron $Ca^{2+}$ imaging in moving animals[59] was modified to allow axonal visualization with an improved temporal accuracy and signal-to-noise ratio. Animals expressing a GCaMP6 indicator exclusively in RIS were imaged for 60 s while moving undisturbed on transparent 1.5% agar pads in M9 buffer. A ×25/0.6 NeoFluar long-range air objective (Zeiss, Germany) was used to visualize the RIS neuron. Fluorescent measurements of GCaMP6 and mCherry were enabled using a GFP/mCherry Dualband ET Filterset (F56-019, AHF Analysentechnik, Germany), combined with 532/18 nm and 625/15 nm emission filters and a 565 long-pass beam splitter (F39-833, F39-624, and F48-567, respectively, all AHF). Tracking the animal's head was established by the PhotoTrack system (Applied Scientific Instrumentation, USA) that automatically repositions the motorized XY stage to keep a bright fluorescent marker in center of the field of view (FOV). By keeping the relative signal strength from each of the four sensors in a four-quadrant photomultiplier tube (PMT) equal, via an analog system, millisecond precision is achieved[59]. An oblique 625 short-pass beam splitter (F38-625, AHF) was inserted in the light path to divert the long red wavelengths to the PMT. Pncx-10::mCherry was expressed in the terminal pharyngeal bulb to allow for robust tracking. To exclude longer wavelengths necessary for behavioral analysis, a 615/20 bandpass filter (F39-616, AHF) was also added in the light path to the PMT, thus improving tracking performance. Behavioral images to deduce the animals' body shape and orientation were obtained under far red illumination with a far-red 740 nm LED filtered with a 690/50 bandpass filter (F47-690, AHF) and magnified with a ×4/0.1 long range objective positioned above the sample, optimized with an additional ×0.5 demagnification tube lens. Data were acquired through the transparent agarose pad with a DCC1545M USB CCD camera (Thorlabs, Newton, NY, USA) controlled via the uc480 ThorCam Software on a separate PC. To exclude the bright blue and yellow excitation light (necessary for calcium imaging), a 610–675 nm bandpass filter was added in the behavioral acquisition light path. Synchronization of both camera's running with 30 ms exposure time and the stage position reporting was done with a TTL start and stop trigger pulse sent by a Lambda SC Smart shutter controller unit (Sutter Instruments, USA). The synchronized combined data of 60 s acquisitions enable behavioral quantification of parameters such as speed, velocity, and acceleration along the midbody axis. This axis was determined by the pharyngeal bulb (kept in the center of the FOV of the behavioral camera due to tracking) and the centroid of the animal's body shape. Prior to differentiation, the stage coordinates of the fluorescent pharyngeal bulb in the moving head were corrected by the deviation of body centroid of the complete animal. The normalized velocity or speed in Fig. 6b, f and Supplementary Fig. 5 were calculated by pairwise subtracting the mean values of the corresponding time periods for individual animals and dividing this by the average speed over the full 60 s acquisition of that animal.

**Image processing**. Video files containing data of both fluorescent channels (for GCaMP6 and mCherry) were processed with custom written Mathematica notebooks (Wolfram Research, Inc., Champaign, IL, USA) as also depicted in Fig. 5b and Supplementary Movie S5. Green and red fluorescent channels were digitally overlaid to accurately correct the spatial alignment. As the structure tracked was not the cell body of RIS but the pharyngeal terminal bulb of a moving animal, the RIS neuron image rotates around the center of the image. A data analysis pipeline was programmed to allow parallelized processing of multiple videos. First, the green fluorescent channel was cropped to around the respective position of the pharyngeal bulb. Next, images were thresholded to locate the centroid of the moving neuronal cell bodies in every frame of the video. The cell body was then repositioned to overlay its position in all frames and rotated after measuring the angular orientation to horizontal of both gut autofluorescence and the brightest regions of the axon (by binarizing a smaller ROI after masking the gut). Subsequently, also the dorso-ventral axis was determined. In this way, an ROI containing the entire RIS morphology was cropped and rotated in each image such that the axon is always oriented in the same direction. At last, a parabola was fitted through the neuronally shaped image components. Alongside this parabola, about 25 equally spaced points were determined to generate rectangular ROIs perpendicular to it. Mean fluorescent intensity was measured in these sub-ROIs (5 × 25 pixels) and subtracted by the median intensity of two adjacent smaller regions above and below each of these (3 × 5 pixels) to correct for local differences in background along the axon. Finally, these values were resampled to 100 values representing the percentagewise segments along the axon to correct for stretching of the axon. The $\Delta F/F$ traces in Fig. 5c, d are the mean values of the 50 most anterior segments of RIS covering the nerve ring region represented as $\Delta F/F = (F_i - F_{mean})/F_{mean}$, where $F_{mean}$ is the mean fluorescent intensity during the first 2 s of acquisition. Fluorescent heatmaps in Fig. 5c, d display min–max normalized values (for each segment of each video normalized over time by $F_{norm,i} = (F_i - F_{min})/(F_{min} - F_{max})$ to account for variation in the mean intensity of each segment) that are smoothed over four frames with a mean filter. Figure 5e displays the mean of 45 aligned normalized fluorescence time windows 6 s prior and post manually assigned $Ca^{2+}$

peak events. The normalized fluorescent traces in Fig. 6b, f are the means ± SEM of event-aligned 6 s time windows of min–max normalized traces that consist of the mean fluorescence intensity of the 50 nerve ring segments.

**Calcium dynamics to behavior correlation analysis**. Conditional probability density functions (in Fig. 6a, e) of the shortest unbiased time lag to either a reversal or a given $Ca^{2+}$ peak event or vice versa were found by Gaussian kernel density estimation with bandwidth 1 s and aligned to the conditional event ($Ca^{2+}$ peak or reversal respectively). Cross-correlation analyses (Fig. 6c, g) were performed in Mathematica by calculating Pearson's correlation functions for equally sized 6 s time windows of both normalized fluorescent change and acceleration time shifted up to 3 s time lags.

**Electrophysiology**. Recordings from dissected BWM cells anterior to the vulva at the ventral side are described in ref. [40]. Light activation was performed using an LED lamp (KSL-70, Rapp OptoElectronic, Hamburg, Germany; 470 nm, 8 mW/mm²) and controlled by an EPC10 amplifier and Patchmaster software (HEKA, Germany). mPSC analysis was done by the Mini Analysis software (Synaptosoft, Decatur, GA, USA, version 6.0.7). Amplitude and mean number of mPSC events per second were analyzed for each 10 s before, 10 s during, and 10 s after illumination.

**Electropharyngeograms**. EPG recording was performed as previously described[69]. Animals were selected on the day prior to measurement as L4 larvae. The head was cut away from the body with a scalpel at about one third to half of the body length. The tip of the worm head was sucked under 100-fold magnification into an EPG-suction electrode, connected via a silver chloride-coated silver wire to an EPC-10 amplifier (Heka, Germany). Prior to measurement, the preparation was incubated in 2 μM serotonin for 5 min to induce pharyngeal pumping. EPG recordings were performed by the PatchMaster software (Heka). We recorded spontaneous pumping for 30 s prior to 30 s of RIS::ChR2 depolarization via 3 mW/mm² (470 nm) light and further 30 s post stimulus. We used the Review software (Bruxton Corporation, USA) to convert from PatchMaster to ABF files. Pump rate and duration were analyzed by AutoEPG58 (kindly provided by C. James, Embody Biosignals Ltd., UK).

**Statistics**. Statistical analysis was performed in Prism (Version 5.01, GraphPad Software, Inc.), Mathematica (version 10, Wolfram Research, Inc., Champaign, IL, USA), OriginPro 2015G (OriginLab Corporation, Northampton, USA), R (version 3.3.2), the latter aided by RStudio (version 1.0.136, RStudio, Inc.), or KNIME (Desktop version 3.5, KNIME.com AG, Zurich, Switzerland). No statistical methods were applied to predetermine sample size. However, sample sizes reported here are consistent to data presented in previous publications. Data were tested for normality prior to statistical inference. Data are given as means ± SEM when not otherwise stated. Significance between data sets after paired or two-tailed Student's $t$ test is given as $p$ value (*$p \leq 0.05$; **$p \leq 0.01$; ***$p \leq 0.001$), when not otherwise stated. For other statistical tests used, see figure legends.

**Reporting summary**. Further information on research design is available in the Nature Research Reporting Summary linked to this article.

## Data availability
All data are provided as Supplementary Information. Reagents and code are available upon request.

## Code availability
Codes are available upon request.

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

## Acknowledgements

We are grateful to N. Pokala, C. Bargmann, O. Tolstenkov, K. Erbguth, and D.S. Kim for reagents and to A. Hermann, J. Weil, L. Gottschlich, L. Anneser, M. Steiner, H. Fettermann, M. Höret, R. Wagner, K. Zehl, and F. Baumbach for expert technical assistance. We thank I. Beets, and members of the Gottschalk and Schoofs laboratories for discussion and comments on the manuscript. Some strains were provided by the *Caenorhabditis* Genetics Center, which is funded by the NIH - Office of Research Infrastructure Programs (grant P40 OD010440) and the National Bioresource Project for the Experimental Animal "Nematode *C. elegans*". This work was funded by Goethe University and by the Deutsche Forschungsgemeinschaft (DFG) grants FOR1279 (GO1011/4-1, 4-2) and GO1011/13-1 to A.G. and EXC115 (Cluster of Excellence Frankfurt) to E.H.K.S. and A.G.

## Author contributions

Conceptualization: W.S.C., P.V.d.A., C.G., S.C.F., A.G.; data curation, analysis, and visualization: W.S.C., P.V.d.A., C.G., J.F.L., C.S., M.B., S.W., A.O., S.C.F., A.G.; software: W.S.C., P.V.d.A., S.C.F.; reagents: F.M., H.B.; wrote the paper: W.S.C., P.V.d.A., A.G., with help from the other authors; supervision and funding acquisition: L.S., E.H.K.S., A.G.

## Additional information

**Competing interests:** The authors declare no competing interests.

