## [Transparent Peer Review File · Nature Communications]

Reviewers' comments:

Reviewer #1 (Remarks to the Author):

This study identifies a single command neuron in the nematode *C. elegans* that initiates a behavioral relevant stop of locomotion. When optogenetically activated the neuron evokes an instantaneous halt of the locomotion with maintained posture. The halt might be followed by a reversal of the locomotor direction. With a battery of genetic perturbations the authors show that both GABA and a peptide released from the command neuron are essential for initiation and maintenance of the stop. Using elegant genetically driven calcium imaging combined with behavioral and electrophysiological recordings the authors show that the halt is due to a halt of oscillation of muscles or motor neurons, and that the command neuron firing is related to the stop with a complicated calcium dynamic in its branches that presumably lead to diverse behavioral outcome.

The study thus identifies an important component in the locomotor control needed for the episodic expression of locomotion in *C. elegans*: namely the ability to stop forward locomotion. Such a stop mechanism has recently been identified both in rodents and fish. The present study underscores the evolutionary importance of this mechanism. The study is elegant and well-carried out. It adds to an increasing number of motor control studies identifying the mechanism for behavioral choices. I am enthusiastic about the study but have some concerns about the presentation as outline below plus additional points that needs to be clarified.

1) A major concern I have about the study is that it is hard to digest the results for a non *C. elegans* person, which makes the study less accessible for the general readership. This is partly due to a somehow convoluted presentation of results with rather complicated and non-intuitive figures. In Figure 1D it is hard to understand what elongation means (needs to be visualized with respect to the body curvature?). In Figure 2 and Figure 3 it is hard to appreciate that the frequency of the muscle and the motor neuron oscillations stop – which is absolutely essential for the story. Panels like those illustrated in suppl Figures 2 and 4 should be included in these main figures. Figure 4 is essential – but there is no definition of all the genetic models and what they mean (egl is not even defined in the main text). Figure 5B makes little sense and should probably be in the supplement. The description around Figure 6 is very confusing and seems even contradictory with respect to flp-11. It appears from Fig. 6F that flp-11 mutants have calcium increase but no slowing yet on line on line 398 it says that RIS activity suffices for induction of slowing both in WT and flp-11 mutants. The entire text around this figure needs careful editing to be more straight forward in reading. Figure 7 and the compartmentalization is also not very easy to understand and needs better explanation. Are the authors indicating that there is compartmentalized transmitter-release at one or both terminal areas of GABA and peptide? And what post-synaptic neurons are found these two areas? Without this knowledge, it seems very hard to understand the results at least for this reader.

2) There is a heavy use of abbreviations or *C. elegans* specific terms. It would be helpful to have them spelled out: RIS, Flp-11, BMW, VB, DB, RMD, SMD; SDQ, Q system, egl, unc31, unc-47 AVJ, PVC, SMD, NPR-2 etc. In many places the text is difficult to follow because of this and because the *C. elegans* specificities are not explained so well

3) The mechanism for how the locomotion is halted is presumably similar to what has been proposed for the mouse 'stop cells': inhibition of rhythm generation without inhibiting the motor neuron output. In the mouse rhythm generation is taking place in interneurons while in *C. elegans* rhythm appears to happen directly among motor neurons (?). How is it possible to inhibit the rhythm without affecting the tonic activity in motor neurons? Are GABA and flt-11 peptide both inhibitory neurotransmitters? Some speculation/explanation would be nice here.

4) The authors suggest that the reversal of locomotion is not directly induced by RIS activity. What else could it be and how can the animal select halt without reversal and halt with reversal given the calcium dynamic in Figure 7?

5) Some discussion of the functional switch of RIS neuron from a sleep-inducing to a stop-inducing

functions seems appropriate.

6) Minor changes:

Line 40: 'rhythm generation, usually by mutually inhibitory networks of neurons that oscillate' this can be misunderstood; should be excitatory neurons that oscillate.

Line 43: Add Talpalar et al. 2013 and Kiehn 2016 as reference to the left-right coordinating system.

Line 44: reference 4 is not correct – better reference for MLR in mammals will be Caggiano et al. 2018 or a review of MLR

Reference 7 is not correct with respect to stop neurons in fish.

Line 98: "with the induction of reversals, and required FLP-11 signaling" ? is that what Figure 7 says.

Reviewer #2 (Remarks to the Author):

General comments

This paper identifies and characterizes a single *C. elegans* neuron that causes the animal to stop moving forward. The neuron has several interesting features such as command-neuron like correlation between activity and behavior, parallel signaling by GABA and a neuropeptide, and compartmentalized calcium signaling in its neurite.

Whereas it is always exciting to see the discovery of a new command-like neuron, the paper has a number of drawbacks in its current form. One problem is that the claim that this is the first stop neuron in an invertebrate appears to be incorrect; Tastekin et al., eLife 2018; 7:e38740 seems to be a direct precedent. Another problem is that although the functional characterization of neuron is impressively thorough and interesting, the authors do not describe general principles of function or neural computation; the paper will keenly interest *C. elegans* systems neuroscientists, but it is unclear how much further its interest will extend. Finally, the paper is difficult to read in certain places, notable examples being the description of the key analysis in Fig. 6A,E, and lines 480-485 in the Discussion.

Specific comments

The authors present ample evidence that neuron, RIS, is sufficient for stopping, but I was unable to find evidence that it is necessary for locomotory stopping in a freely moving animal, the second defining characteristic of a command neuron. Nor do the authors provide a technical explanation for why this key experiment was not done. The persistence of reversals in the absence of flp-11, noted in the Discussion, partially addresses this concern, but there remains the interesting question of whether RIS is required for the normal frequency or duration of stops.

In Fig. 2B the authors use calcium imaging to show that oscillations in body-wall muscle are terminated by photo-activation of RIS. However, the fact that these oscillations are not traveling waves weakens the interpretation that RIS terminates locomotion, which is based on traveling waves. A related problem is that the oscillation period is 0.1 Hz, or about 4-fold slower than in freely-moving worms. The reasons for and the consequences of each of the issues should have been discussed.

In Fig. 2C, middle panel, there is no second positive peak in the autocorrelation. Therefore, the authors' stipulation that in such cases the oscillation period is 20 sec seems simply wrong – there may be no oscillation whatsoever. Indeed this seems to be the case.

In Fig. 3, muscle electrophysiological recordings are said to have been made "anterior to the vulva." This description is vague, leaving open the possibility that the recordings were ANYWHERE anterior to the vulva. This is a concern because Fig. 2B suggests that oscillations occur only in BWM in a narrow region midway between head and tail; that then, is where the recordings should have been made, exclusively.

Page 9, line 215. The head motor neurons are said to oscillate at 1/6 Hz. Why are there no corresponding oscillations in body wall muscles of the head? Why are these oscillations so different from the 0.1 Hz oscillation of body wall muscles in the midbody?

The authors imply that *unc-31* is required only for the secretion of neuropeptides whereas in fact it is also required for normal secretion of other transmitter/modulators such as biogenic amines and endocannabinoids. How might this affect the interpretation of the mutants?

Fig. 5C,D,E: please label cell body and neurite.

Page 17, line 398. "In sum, locomotion is miss-regulated in *flp-11* mutants, yet RIS activity suffices for induction of slowing in both WT and *flp-11* mutants." This statement seems to contradict Fig. 6F, which shows no slowing in *flp-11* mutants.

Page 21, line 480 "PVC innervates RIS and RIS has synaptic output to AVE55, implying a possible sequence of signaling leading from forward (PVC) to a pause (RIS) and reversal (AVE) transition. During pure slowing events, branch Ca^{2+} activity was suppressed prior to RIS activation, suggesting that this sequence of events may underlie the suppression of reversals. Possibly, PVC activity, which may be inhibitory to RIS (e.g. via mAChRs, as PVC is cholinergic), modulates RIS output, altering the evoked behavior from a reversal to a stop." This scenario doesn't seem to make sense. The proposed pathway (PVC-RIS-AVE) entails two inhibitory connections in series, which amounts to an excitatory connection. That means that a transient excitation of PVC (unlikely in itself), producing the blue dot in Fig. 7B, would produce a transient disinhibition of AVE, making reversals more likely, not less likely.

Reviewer #3 (Remarks to the Author):

The major components of *C. elegans* navigation are forward motion, reversals, and turns. Transitions between forward and reverse involve slowing and stopping, but whether there is a specific neural mechanism regulating a "stop" state, as predicted by some models, has not been clear. Costa et al. show that optogenetic activation of a single interneuron, RIS caused immediate stopping without loss of posture, as with flaccid paralysis, and is associated with loss of oscillatory Ca^{++} activity in body wall muscles. This depended on the peptide FLP-11. The RIS axon exhibits compartmentalized Ca^{++} dynamics, and these show different activity patterns between stops that are or are not associated with reversals. RIS has a previously described role during developmentally-timed lethargus...this work suggests in adulthood it functions in locomotion to regulate state transitions.

This is an excellent paper that uses interesting and innovative approaches. I think there is some analysis that could be done with data in hand that might make the findings clearer.

The one major weakness is that there are no cell-specific loss-of-function experiments with respect to RIS. Optogenetic stimulation is informative as a partial test of sufficiency, but is also likely represents a non-physiological form of gain-of-function activity. Likewise, coupling stimulation with *flp-11* mutation is an important result, but *flp-11* is expressed in many other neurons,

complicating interpretation. With so many tools to cell-specifically interfere with function/survival of genetically addressable cells in *C. elegans* (ablation, tetanus toxin, opto/chemogenetic inhibitors, caspases), this feels like a gap in the analysis.

Comments

- The autocorrelation function in Fig 2C (“During”) appears to show no periodicity at all....Is there a reason for choosing 20s other than it being 2x the 10s stimulation time? If here and in Fig3 all aperiodic cases get essentially “binned” at 20s because of arbitrary experimental parameters, I am not sure this is an appropriate input to the kinds of statistical tests used in Fig 3G. I understand the challenge of a quantitative comparison of time series, and the result seems clear....perhaps comparisons of peak values in the frequency domain would avoid this problem?
- Fig 4. It is interesting that mutants (*flp-11* and *egl-3*) that have reduced stopping when RIS is stimulated all show increased reversals. Why should this be the case? Does it tell us something about the circuits coordinating the transition from forward to reverse locomotion?
- Fig 4. Because both are used to test for the involvement of peptidergic signaling, the different effect of *unc-3* and *unc-31* warrants some discussion
- Fig 5C it would be helpful to have some labels of landmarks to ease reading the heat map (soma, axon, branch) for the green fluorescence images.
- Fig 5C...there is a visible periodic signal mostly in the negative (blue/white) range of the heat maps... is this motion? It appears approximately the frequency of locomotion...if so, is it an imaging artifact, or perhaps related to RIS connectivity with SMDs?
- The circuit connectivity of RIS and its relation to synaptic connectivity in different compartments is only briefly discussed, despite this likely being critical to RIS function. Is the branch essentially part of a separate circuit, or is it global excitation that is sometimes coupled with local inhibition? The representative heatmaps in Fig 4, particularly 4C, are suggestive of further compartmentalization in the nerve ring (e.g. signals in only a portion of the nerve ring at 22s and 30s, compared to signals throughout at 4s, 12s, 45s). Like RIA, RIS appears to have separable classes of calcium events that perform a sensorimotor gating function. RIS expresses a number of ionotropic and metabotropic receptors that might underlie these different Ca^{++} events...were any of these mutants tested? If the authors did it would be informative to include even negative results from these experiments.
- A more thorough analysis of how different calcium events within RIS relate to each other (not just to behavior) could be performed on data already in hand. For example, does branch activity (increased or decrease Ca^{++}) ever occur uncoupled from axonal events, or there really just two classes of event, axon + branch and axon preceded by branch inhibition?

Minor suggestions/comments

- I'd like to think we are past the “sleep-like state” language around sleep in invertebrates...certain *C. elegans* quiescent states clearly fit the definition of sleep in animals, and the only reason to hedge is to avoid offending the delicate sensibilities of people who only think about mammals.
- Fig 1. sample sizes should be reported for each group being compared
- Fig 1e – why one box plot and points? Matched pairs may be best shown with lines showing paired comparison plot to be consistent with the statistical test used (as in 2D)
- There seems to be something unusual with the normalization of Ca^{++} traces in Fig5C and 5D—normalization is if to F0 but traces begin negative. .
- The diverging colormap used in Fig 5 heatmaps should either have white = 0 (so cool = negative and warm = positive) or a non-diverging colormap should be used. I'd suggest the latter since 0 is arbitrary.
- Line 397 “miss-regulated”

Reviewers' comments:

Reviewer #1 (Remarks to the Author):

This study identifies a single command neuron in the nematode *C. elegans* that initiates a behaviorally relevant stop of locomotion. When optogenetically activated the neuron evokes an instantaneous halt of the locomotion with maintained posture. The halt might be followed by a reversal of the locomotor direction. With a battery of genetic perturbations the authors show that both GABA and a peptide released from the command neuron are essential for initiation and maintenance of the stop. Using elegant genetically driven calcium imaging combined with behavioral and electrophysiological recordings the authors show that the halt is due to a halt of oscillation of muscles or motor neurons, and that the command neuron firing is related to the stop with a complicated calcium dynamic in its branches that presumably lead to diverse behavioral outcomes.

The study thus identifies an important component in the locomotor control needed for the episodic expression of locomotion in *C. elegans*: namely the ability to stop forward locomotion. Such a stop mechanism has recently been identified both in rodents and fish. The present study underscores the evolutionary importance of this mechanism. The study is elegant and well-carried out. It adds to an increasing number of motor control studies identifying the mechanism for behavioral choices. I am enthusiastic about the study but have some concerns about the presentation as outlined below plus additional points that need to be clarified.

1) A major concern I have about the study is that it is hard to digest the results for a non-*C. elegans* person, which makes the study less accessible for the general readership. This is partly due to a somewhat convoluted presentation of results with rather complicated and non-intuitive figures. In Figure 1D it is hard to understand what elongation means (needs to be visualized with respect to the body curvature?).

We thank the reviewer for making us aware of this weakness. We rewrote and rearranged the text of the manuscript, and took more care in describing *C. elegans*-specific nomenclature, etc. We also described the figures better and added explanatory pictograms, for example to better illustrate the representation of body elongation in figure 1D.

In Figure 2 and Figure 3 it is hard to appreciate that the frequency of the muscle and the motor neuron oscillations stop – which is absolutely essential for the story. Panels like those illustrated in supplementary Figures 2 and 4 should be included in these main figures.

We included the panels from the supplementary figures.

Figure 4 is essential – but there is no definition of all the genetic models and what they mean (*egl* is not even defined in the main text).

We added more precise descriptions of the genes and pathways that act in RIS, as part of the caption above each panel in Fig. 4A, and we described the abbreviations of gene names in the text.

Figure 5B makes little sense and should probably be in the supplement.

We removed half of the figure to Supplements (now S4), but left the lower half, as we thought it helps to clarify the procedure of our image analysis.

The description around Figure 6 is very confusing and seems even contradictory with respect to flp-11. It appears from Fig. 6F that flp-11 mutants have calcium increase but no slowing yet on line on line 398 it says that RIS activity suffices for induction of slowing both in WT and flp-11 mutants. The entire text around this figure needs careful editing to be more straight forward in reading.

Our apologies, we now rewrote and disentangled this section, also made it more concise to lead the reader to our main conclusions more easily. We also took care to remove inconsistencies from our text.

Figure 7 and the compartmentalization is also not very easy to understand and needs better explanation.

We re-did this section, too. The figure is complex because we had to describe eight ‘degrees of freedom’:

- 2 Genotypes
- 2 Behaviors
- Time domain
- Relative position in neuron
- Calcium signal
- Significance of change in Ca^{2+} in relation to start of event

The first two are given by the 2x2 plot, time and position are the x,y axis, color is Ca^{2+} signal and significance was reduced by either plotting or not plotting the color code.

Are the authors indicating that there is compartmentalized transmitter-release at one or both terminal areas of GABA and peptide?

No, this we cannot conclude or state, we only observed compartmentalized Ca^{2+} dynamics in the branch and the nerve ring process. At stage, it is not known where GABA and neuropeptide are released and whether this occurs in distinct sites, or whether it is co-release. Above is a sketch from the famous wiring diagram paper of John White, which shows the few synaptic and gap junction contacts in the branch. The branch is post-synaptic to the two PVC neurons as well as AVJ (arrows), and forms gap junctions with AVJ, and the head motor neuron SMDVL (rotated ‘T’s).

And what post-synaptic neurons are found these two areas? Without this knowledge, it seems very hard to understand the results at least for this reader.

We made an effort to describe this better. We also added a figure (now S3A, B), depicting the connectome of RIS, and discuss its possible implications in the light of our data.

2) There is a heavy use of abbreviations or C. elegans specific terms. It would be helpful to have them spelled out: RIS, Flp-11, BMW, VB, DB, RMD, SMD; SDQ, Q system, egl, unc31, unc-47 AVJ, PVC, SMD, NPR-2 etc.

In several cases, we now describe what these abbreviations stand for, though this is often not too helpful. E.g. egl means ‘egg-laying defective’. The names of neurons are often arbitrary, or relate to

their localization in the body, like RIS = ‘ring interneuron S’. Thus, to spell out all these abbreviations is not really useful, but we did this where it makes sense or helps the readability.

In many places the text is difficult to follow because of this and because the *C. elegans* specificities are not explained so well

We tried our best to improve this throughout the manuscript.

3) The mechanism for how the locomotion is halted is presumably similar to what has been proposed for the mouse ‘stop cells’: inhibition of rhythm generation without inhibiting the motor neuron output. In the mouse rhythm generation is taking place in interneurons while in *C. elegans* rhythm appears to happen directly among motor neurons (?). How is it possible to inhibit the rhythm without affecting the tonic activity in motor neurons?

We don’t think, the tonic activity of motor neurons (i.e. a rise in their membrane potential) is inhibited, as we should have seen an overall decrease in mini frequencies. Only the Fourier transform analysis showed that the low frequencies become less populated during RIS activation. Possibly, less temporal coherence in muscle input thus leads to lower temporal coordination also of muscle action potentials, leading to reduced oscillatory muscle activity. As there is no coordination of muscular activity across the animal anymore, the locomotion stops.

Are GABA and flt-11 peptide both inhibitory neurotransmitters? Some speculation/explanation would be nice here.

GABA is mostly inhibitory, though there are excitatory GABA receptors in *C. elegans*. It is not known if such receptors are expressed downstream of RIS pre-synapses. Also, FLP-11 peptides have been shown in *Ascaris*, a large nematode highly homologous to *C. elegans*, to act inhibitory, actually also directly to muscle (we mention this and added a reference to the paper text). The exact mechanism would require to determine the identity of the FLP-11 receptors, and their expression patterns. We now discuss these points in the manuscript.

4) The authors suggest that the reversal of locomotion is not directly induced by RIS activity. What else could it be and how can the animal select halt without reversal and halt with reversal given the calcium dynamic in Figure 7?

RIS likely acts in concert with other neurons. Depending which ones these are, the behavior may be different, and RIS contributes to this by mediating the slowing response needed for both types of behavior, slowing and slowing with concomitant reversal. As in FLP-11 mutants, no stopping occurred, yet reversals were found, this might hint at a disturbed balance within the neuronal network. To illustrate this, we included the RIS connectome figure (S3A), and discuss possible pathways, in the light of the literature. We also provide new data about the gap junction subunit UNC-9, that should underlie the connection between RIS and the RIM neuron, which affects reversals, and about the tyramine transmitter used by the RIM neuron. This data is not part of Fig. 4 and provides some clues to the question of the reviewer. Yet, it is clear that definitive answers will require (much) more work, with specific knock-down of candidate receptors (as yet unknown) in distinct cells, and localization of release sites for GABA and FLP-11 in the branch vs. the nerve ring process. The fact that reversals occur after the end of RIS photostimulation in wild type animals may also hint at some rebound activation when the inhibitory signal from RIS ends.

5) Some discussion of the functional switch of RIS neuron from a sleep-inducing to a stop-inducing functions seems appropriate.

We have altered this conclusion, in the light of recent findings of the lab of Henrik Bringmann (Max Planck Institute, Göttingen, and Marburg University), who provided helpful reagents and is now also a co-author. The Bringmann lab showed that stress-induced sleep is found in adult animals, and also involves RIS. However, the timescales on which RIS acts in sleep control (minutes to hours) and the times it is active in locomotion control (seconds) may represent two extremes of a continuum of activity durations, depending of RIS’ dual uses in the compact *C. elegans* nervous system. We now altered the title and our discussion this way, and toned down the idea of a developmental switch.

6) Minor changes:

Line 40: ‘rhythm generation, usually by mutually inhibitory networks of neurons that oscillate’ this can be misunderstood;

should be excitatory neurons that oscillate.

Line 43: Add Talpalar et al. 2013 and Kiehn 2016 as reference to the left-right coordinating system.

Lien 44: reference 4 is not correct – better reference for MLR in mammals will be Caggiano et al. 2018 or a review of MLR
Reference 7 is not correct with respect to stop neurons in fish.

Line 98: “with the induction of reversals, and required FLP-11 signaling” ? is that what Figure 7 says.

We thank the reviewer for these suggestions and implemented these changes. We clarified the description and discussion of Fig. 7.

Reviewer #2 (Remarks to the Author):

General comments

This paper identifies and characterizes a single *C. elegans* neuron that causes the animal to stop moving forward. The neuron has several interesting features such as command-neuron like correlation between activity and behavior, parallel signaling by GABA and a neuropeptide, and compartmentalized calcium signaling in its neurite.

Whereas it is always exciting to see the discovery of a new command-like neuron, the paper has a number of drawbacks in its current form. One problem is that the claim that this is the first stop neuron in an invertebrate appears to be incorrect; Tastekin et al., eLife 2018;7:e38740 seems to be a direct precedent.

We thank the reviewer for making us aware of this paper, which came out in the final stages of our initial submission. We now included this reference and briefly cover its findings.

Another problem is that although the functional characterization of neuron is impressively thorough and interesting, the authors do not describe general principles of function or neural computation; the paper will keenly interest *C. elegans* systems neuroscientists, but it is unclear how much further its interest will extend.

We expanded on this in the revised manuscript, e.g. by including now coverage of the connectome up- and downstream of RIS. We did this to the extent that we can, in the light of our data, without over-interpretation. Deducing more general principles of computation would require also electrophysiological recording of neuronal activity downstream of RIS, which is extremely challenging given the small size of the animal, but also the limited information available on the transmitter receptors that may be involved here (see also above, reply to reviewer #1). However, we also provide new data about the gap junction subunit UNC-9, that should underlie the connection between RIS and the RIM neuron, that affects reversals, and about the tyramine transmitter used by the RIM neuron. This data is not part of Fig. 4.

Finally, the paper is difficult to read in certain places, notable examples being the description of the key analysis in Fig. 6A,E, and lines 480-485 in the Discussion.

We worked on this carefully and hope that these sections are now clearer.

Specific comments

The authors present ample evidence that neuron, RIS, is sufficient for stopping, but I was unable to find evidence that it is necessary for locomotory stopping in a freely moving animal, the second defining characteristic of a command neuron. Nor do the authors provide a technical explanation for why this key experiment was not done.

The reason we did not include such data was that we did not succeed in generating the respective strains or did not observe experimental effects. We had tried: Expression of halorhodopsin (no behavioral effects observed), expression of miniSOG (no light-induced cell killing observed), Expression of histamine-gated Chloride channel (HisCl – no effects observed), expression of tetanus toxin (no transgenic worms could be generated). Now, we joined forces with Henrik Bringmann (see above), who had generated animals in which RIS was genetically ablated. We analyzed locomotion in these animals and observed a significant reduction in the occurrence of reversals and stops in the RIS-ablated animals. This data is now presented in Fig. 1F.

The persistence of reversals in the absence of flp-11, noted in the Discussion, partially addresses this concern, but there remains the interesting question of whether RIS is required for the normal frequency or duration of stops.

We now provide this data.

In Fig. 2B the authors use calcium imaging to show that oscillations in body-wall muscle are terminated by photo-activation of RIS. However, the fact that these oscillations are not traveling waves weakens the interpretation that RIS terminates locomotion, which is based on traveling waves.

These waves are not traveling because of the immobilization. It is only observed in a moving animal, as it is entrained by and requires the proprioceptive feedback, that is absent in an immobilized animal. With our tracking system, we cannot follow muscle calcium in a moving animal. However, the fact that locomotion requires oscillating activity in muscle, and that RIS stops these oscillations, is a very strong indication that this also underlies the stop induced by RIS activation in moving animals.

A related problem is that the oscillation period is 0.1 Hz, or about 4-fold slower than in freely-moving worms. The reasons for and the consequences of each of the issues should have been discussed.

This is, as described above, due to the immobilization and lack of proprioception. It has been described by several people in previous publications. This is now discussed briefly in line 151.

In Fig. 2C, middle panel, there is no second positive peak in the autocorrelation. Therefore, the authors' stipulation that in such cases the oscillation period is 20 sec seems simply wrong – there may be no oscillation whatsoever. Indeed this seems to be the case.

We also think that there are no oscillations. We have not described this as good as we should have, thank you for pointing this out. In these experiments, we used a relatively short stimulation period. To allow estimating the reduction of oscillations, we assumed that the period would be at least 20 s. We now changed the y-axis label ('lowest assignable period') to reflect this fact, and to not give the impression that the data in the presence of stimulation is unambiguous.

In Fig. 3, muscle electrophysiological recordings are said to have been made "anterior to the vulva." This description is vague, leaving open the possibility that the recordings were ANYWHERE anterior to the vulva. This is a concern because Fig. 2B suggests that oscillations occur only in BWM in a narrow region midway between head and tail; that then, is where the recordings should have been made, exclusively.

The notion of oscillations only halfway to the head is due to the impression given by the example we picked for the figure and associated video. Static proprioceptive feedback from the neighboring segments causes reduced muscle Ca^{2+} levels in the regions of low curvature, while high curvature induces high basal activity and Ca^{2+} level in the muscle nearby, due to proprioceptive activation of the motor neurons. Thus, with other curvature 'frozen' in the immobilized animal, oscillations were also clearly visible in other regions. We could not detect them in the head, due to the very bright fluorescence of the pharynx in these animals, expressing mCherry as a marker for transgenic animals, which interferes with RCaMP imaging. In any case, the muscle cells that were patch clamped are always very close to / anterior of the vulva, i.e. precisely the region where the strong oscillations are seen in our example video. These positions are indicated in the sketch below (red muscle cells were patched; anterior is up):

Page 9, line 215. The head motor neurons are said to oscillate at 1/6 Hz. Why are there no corresponding oscillations in body wall muscles of the head? Why are these oscillations so different from the 0.1 Hz oscillation of body wall muscles in the midbody?

See above, we could not image muscle Ca^{2+} in the head, so the oscillations may be the same as those of the head neurons. However, the neurons are not connected 1:1 to a given muscle, and we don't know the identity of the neurons we image. So it is difficult to correlate the activities. Yet, during normal locomotion, the head oscillations are usually twice as frequent as the body wave, since the animal samples the environment with its nose. So this is consistent with the observed faster oscillations in the head neurons.

The authors imply that *unc-31* is required only for the secretion of neuropeptides whereas in fact it is also required for normal secretion of other transmitter/modulators such as biogenic amines and endocannabinoids. How might this affect the interpretation of the mutants?

This is possible, however, we are not aware of data showing this in *C. elegans*. Should we have overlooked this, please let us know. In any case, we used the *egl-3* pro-protein convertase mutant in this experiment to conclude that neuropeptidergic signaling is required for RIS effects, not the *unc-31* data. We also assessed *unc-31*, as it is often used for the same purpose, and indeed affects numerous neuropeptides, and as it was used also in previous sleep-related experiments characterizing RIS.

Fig. 5C,D,E: please label cell body and neurite.

Done, we added a pictogram to clarify this.

Page 17, line 398. "In sum, locomotion is miss-regulated in *flp-11* mutants, yet RIS activity suffices for induction of slowing in both WT and *flp-11* mutants." This statement seems to contradict Fig. 6F, which shows no slowing in *flp-11* mutants.

We clarified this sentence.

Page 21, line 480 "PVC innervates RIS and RIS has synaptic output to AVE55, implying a possible sequence of signaling leading from forward (PVC) to a pause (RIS) and reversal (AVE) transition. During pure slowing events, branch Ca^{2+}

activity was suppressed prior to RIS activation, suggesting that this sequence of events may underlie the suppression of reversals. Possibly, PVC activity, which may be inhibitory to RIS (e.g. via mAChRs, as PVC is cholinergic), modulates RIS output, altering the evoked behavior from a reversal to a stop." This scenario doesn't seem to make sense. The proposed pathway (PVC-RIS-AVE) entails two inhibitory connections in series, which amounts to an excitatory connection. That means that a transient excitation of PVC (unlikely in itself), producing the blue dot in Fig. 7B, would produce a transient disinhibition of AVE, making reversals more likely, not less likely.

We re-wrote and expanded this section and added also a figure showing the RIS connectome. It becomes clear, however, that without distinct experiments, probing elements of this connectome (cells, transmitters, receptors), it is not possible to suggest a single, unifying model. So we toned this down.

Reviewer #3 (Remarks to the Author):

The major components of *C. elegans* navigation are forward motion, reversals, and turns. Transitions between forward and reverse involve slowing and stopping, but whether there is a specific neural mechanism regulating a "stop" state, as predicted by some models, has not been clear. Costa et al. show that optogenetic activation of a single interneuron, RIS caused immediate stopping without loss of posture, as with flaccid paralysis, and is associated with loss of oscillatory Ca^{++} activity in body wall muscles. This depended on the peptide FLP-11. The RIS axon exhibits compartmentalized Ca^{++} dynamics, and these show different activity patterns between stops that are or are not associated with reversals. RIS has a previously described role during developmentally-timed lethargus...this work suggests in adulthood it functions in locomotion to regulate state transitions.

This is an excellent paper that uses interesting and innovative approaches. I think there is some analysis that could be done with data in hand that might make the findings clearer.

The one major weakness is that there are no cell-specific loss-of-function experiments with respect to RIS. Optogenetic stimulation is informative as a partial test of sufficiency, but is also likely represents a non-physiological form of gain-of-function activity. Likewise, coupling stimulation with flp-11 mutation is an important result, but flp-11 is expressed in many other neurons, complicating interpretation. With so many tools to cell-specifically interfere with function/survival of genetically addressable cells in *C. elegans* (ablation, tetanus toxin, opto/chemogenetic inhibitors, caspases), this feels like a gap in the analysis.

See above, similar request of reviewer #2, and our provision of data showing a requirement of RIS for spontaneous stops and reversal behavior.

Comments

- The autocorrelation function in Fig 2C ("During") appears to show no periodicity at all....Is there a reason for choosing 20s other than it being 2x the 10s stimulation time? If here and in Fig3 all aperiodic cases get essentially "binned" at 20s because of arbitrary experimental parameters, I am not sure this is an appropriate input to the kinds of statistical tests used in Fig 3G. I understand the challenge of a quantitative comparison of time series, and the result seems clear....perhaps comparisons of peak values in the frequency domain would avoid this problem?

We have altered our description of the figure, to avoid the impression that the period during light stimulation could be precisely determined. See also reply above to comment of reviewer #2. Also, the labeling of the time scale was wrong, which may have led to further concerns regarding this figure, and which we now corrected.

- Fig 4. It is interesting that mutants (flp-11 and egl-3) that have reduced stopping when RIS is stimulated all show increased reversals. Why should this be the case? Does it tell us something about the circuits coordinating the transition from forward to reverse locomotion?

As discussed above, the reason for this behavior will be explainable from the connectome and a thorough analysis of receptors and transmitters used. We added a figure about the connectome that eases a bit this discussion, but also makes it clear that there is no single answer. Some threads are the RIM interneuron, that should be inhibited to cause reversals, but dual innervation of RIM by RIS (electrical, chemical) make it difficult to interpret this without experiments that probe receptors in RIM and the nature of the gap junctions. We discuss this in more detail in the paper. As described above, we also added data about the gap junction subunit UNC-9, that should underlie this connection, and about the tyramine transmitter used by the RIM neuron. This data is now part of Fig. 4.

- Fig 4. Because both are used to test for the involvement of peptidergic signaling, the different effect of *unc-3* and *unc-31* warrants some discussion

As written above, this has to do with the fact that *egl-3* eliminates all peptide processing, while *unc-31* has been shown to affect peptide release in many but not in all cases. We state this in the paper now (lines 231-234).

- Fig 5C it would be helpful to have some labels of landmarks to ease reading the heat map (soma, axon, branch) for the green fluorescence images.

We did this, thank you for the suggestion.

- Fig 5C...there is a visible periodic signal mostly in the negative (blue/white) range of the heat maps... is this motion? It appears approximately the frequency of locomotion...if so, is it an imaging artifact, or perhaps related to RIS connectivity with SMDs?

This is motion, since the gut expresses some GCaMP as well, and because the RIS cell body moves closer to, and away from the gut in a periodic fashion when the worm moves. We did not include this in our analysis in Fig. 6, thus analyzing only data from the nerve ring portion. The branch data is mostly unaffected by this, as the branch signals were much higher than the motion artefacts. The color scheme we chose was to better allow visualizing the nerve ring and branch signals.

- The circuit connectivity of RIS and its relation to synaptic connectivity in different compartments is only briefly discussed, despite this likely being critical to RIS function. Is the branch essentially part of a separate circuit, or is it global excitation that is sometimes coupled with local inhibition? The representative heatmaps in Fig 4, particularly 4C, are suggestive of further compartmentalization in the nerve ring (e.g. signals in only a portion of the nerve ring at 22s and 30s, compared to signals throughout at 4s, 12s, 45s).

We assume the reviewer meant Fig. 5? We think this is an exciting possibility and would like to explore this further, however, we don't think it is possible with our current set up / in moving animals, due to the deformation of the head and additional motion in the moving animal. We think that this precludes further specific analysis of sub-portions of the nerve ring, as was for example done for the RIA neuron (work by Yun Zhang's lab, doi:10.1038/nature11081), who immobilized the animal and only permitted the head to move dorso-ventrally. We do not think that the quality of our videos, after extraction and processing, allows to draw more conclusions than we suggested, based on our analyses. This might become clear from a new video we provide, (now Supplementary Video 7), in which the deformations of the process are visible. We think they allow to analyze the nerve ring portion as a whole, and the branch as a different entity, but not sub-portions of the nerve ring, since it undergoes a lot of bending.

Like RIA, RIS appears to have separable classes of calcium events that perform a sensorimotor gating function. RIS expresses a number of ionotropic and metabotropic receptors that might underly these different Ca⁺⁺ events... were any of these mutants tested? If the authors did it would be informative to include even negative results from these experiments.

No, we did not. Actually, we had generated an mRNA profile of RIS together with David Miller, Vanderbilt University, and tested some of the candidates from this data set (mostly putative neuropeptide receptors). Only recently, work done by the Shendure lab on single cell transcriptomics, and from our collaborator Henrik Bringmann, made us doubt the 'cleanliness' of our profile. We thus did not want to further embark on this aspect, but would like to do so in the future, in collaboration with Henrik Bringmann.

- A more thorough analysis of how different calcium events within RIS relate to each other (not just to behavior) could be performed on data already in hand. For example, does branch activity (increased or decrease Ca⁺⁺) ever occur uncoupled from axonal events, or there really just two classes of event, axon + branch and axon preceded by branch inhibition?

As above, this is an exciting prospect, however, we fear that the current experimental approach to analyzing axonal Ca²⁺ dynamics are not of sufficiently high quality to permit these analyses.

Minor suggestions/comments

- I'd like to think we are past the "sleep-like state" language around sleep in invertebrates...certain *C. elegans* quiescent states clearly fit the definition of sleep in animals, and the only reason to hedge is to avoid offending the delicate sensibilities of people who only think about mammals.

Yes, ok, we now left out the vagueness about calling it sleep or not.

- Fig 1. sample sizes should be reported for each group being compared

They were in B, but not in D, E, which we now included.

- Fig 1e – why one box plot and points?

Matched pairs may be best shown with lines showing paired comparison plot to be consistent with the statistical test used (as in 2D)

The points are the outliers, the box was shrunk to a seemingly single line.

We now replaced this with a paired comparison plot, however

- There seems to be something unusual with the normalization of Ca^{++} traces in Fig5C and 5D—normalization is if to F0 but traces begin negative. .

This was adjusted. Thank you for spotting this.

- The diverging colormap used in Fig 5 heatmaps should either have white = 0 (so cool = negative and warm = positive) or a non-diverging colormap should be used. I'd suggest the latter since 0 is arbitrary.

We scaled these now to be between 0 and 1. We tried different color maps, however, non-diverging colormaps did not represent the data well / important features were hard to see then.

- Line 397 "miss-regulated"

Corrected.

REVIEWERS' COMMENTS:

Reviewer #1 (Remarks to the Author):

The authors have done a great job in revising their manuscript which has addressed many of the concerns raised by the reviewers. In particular that have reanalysed data and added loss of function data. The revision has also made the presentation more streamlined. I have no further comments.

Reviewer #2 (Remarks to the Author):

The authors have satisfactorily addressed all most of my previous concerns. This includes the acquisition of new data, now shown in Fig. 1F, to address the key question of whether RIS is required for stopping in freely moving animals.

However, I'm still struggling with the problem that the paper may be of limited interest to non-C. elegans researchers. In my previous review I pointed to the lack of elucidation of general principles. Here, I accept the author's response that this is not always possible. But I still think something must be done to attract the interest of a broader readership, most likely by significant revision of the introduction and discussion. The introduction provides a nice summary of what is known about locomotory stop neurons, and sleep neurons, but seems not to identify key questions remaining in this domain. The discussion reads a bit like a laundry list of C. elegans-specific issues addressed by the new research. Perhaps it would help to systematically address the similarities and differences between the circuit mechanisms for stopping in C. elegans, Drosophila, and mammals, with an emphasis, where possible, on the relationship between the internal structure of these mechanisms and the distinct functionality stopping provides in each species.

Minor point: RIS-induced stopping requires the neuropeptide FLP-11, presumably release by dense-core vesicles, which requires unc-31. The manuscript seems to require an explanation for the fact that RIS-induced stopping is essentially unaffected by unc-31 mutations.

Reviewer #3 (Remarks to the Author):

The authors have done a thorough job of addressing my comments, and I remain enthusiastic about the paper and its findings, as detailed in my prior review. I have no further concerns or suggestions. It will be well-received and exciting for the field.

Michael Hendricks

REVIEWERS' COMMENTS:

Reviewer #1 (Remarks to the Author):

The authors have done a great job in revising their manuscript which has addressed many of the concerns raised by the reviewers. In particular that have reanalysed data and added loss of function data. The revision has also made the presentation more streamlined. I have no further comments.

Thank you!

Reviewer #2 (Remarks to the Author):

The authors have satisfactorily addressed all most of my previous concerns. This includes the acquisition of new data, now shown in Fig. 1F, to address the key question of whether RIS is required for stopping in freely moving animals.

Thank you!

However, I'm still struggling with the problem that the paper may be of limited interest to non-*C. elegans* researchers. In my previous review I pointed to the lack of elucidation of general principles. Here, I accept the author's response that this is not always possible. But I still think something must be done to attract the interest of a broader readership, most likely by significant revision of the introduction and discussion. The introduction provides a nice summary of what is known about locomotory stop neurons, and sleep neurons, but seems not to identify key questions remaining in this domain. The discussion reads a bit like a laundry list of *C. elegans*-specific issues addressed by the new research. Perhaps it would help to systematically address the similarities and differences between the circuit mechanisms for stopping in *C. elegans*, *Drosophila*, and mammals, with an emphasis, where possible, on the relationship between the internal structure of these mechanisms and

the distinct functionality stopping provides in each species.

We thank the reviewer for this suggestion. We have now phrased the open question(s) we wanted to address more explicitly in the introduction, and shortened the discussion in some parts (discussion of RIS effects on the pharynx, and the putative FLP-11 peptide receptors) and added text, as well as a figure panel 8C, to give a more general conclusion of our work with respect to other systems.

The suggestion of the reviewer, to “systematically address the similarities and differences between the circuit mechanisms for stopping in *C. elegans*, *Drosophila*, and mammals, with an emphasis, where possible, on the relationship between the internal structure of these mechanisms and the distinct functionality stopping provides in each species” is great, however, we feel we would have to add a lot of text to do this, like in a review article.

Thus, instead, in the new figure, we tried to compare different model systems (worm, fly, tadpole, leech, fish, lamprey, mouse) for current knowledge about stop cells and sleep neurons/systems. We propose that *C. elegans* combines both functions in a single neuron, RIS, distinguished by short or prolonged activity, while more complex brains have dissociated the two functions into distinct cells and systems. From this overview, it also becomes clear that during evolution, many neurotransmitters were employed in these systems, as if a primordial, simple, combined neuron for stop and sleep, evolved into different systems in different branches of the phylogenetic tree, where different molecular / cellular solutions were found for analogous functions. Due to the lack of space / word limit, we remain brief and somewhat superficial, but hope to give enough detail and references in the caption of this figure, to allow readers to follow our argument.

Minor point: RIS-induced stopping requires the neuropeptide FLP-11, presumably release by dense-core vesicles, which requires *unc-31*. The manuscript seems to require an explanation for the fact that RIS-induced stopping is essentially unaffected by *unc-31* mutations.

UNC-31/CAPS is one of the major dense core vesicle (DCV) priming factors for exocytosis. The *unc-31(n1304)* null allele mutation disrupts the 3' splice acceptor of intron 10. Therefore, no UNC-31 protein is formed which leads to severe phenotypic defects. However, bioactive FLP-11 neuropeptides are still generated normally in *unc-31* mutants and are present in DCVs. *unc-31* mutations drastically reduce DCV exocytosis, but do not abolish it completely. If some residual DCV exocytosis persists in *unc-31* mutants, then we would assume that strong photodepolarization of RIS with channelrhodopsin-2 could still induce minimal release of FLP-11 neuropeptides, and potentially enough to sustain the stop phenotype. Alternatively, photodepolarization of RIS with channelrhodopsin-2 might somehow also activate the PKA pathway that bypasses the requirement for UNC-31 in exocytosis. In contrast, *egl-3* mutants that are defective in neuropeptide processing, do only produce and release inactive proprotein peptide precursors (instead of fully mature, amidated FLP-11 neuropeptides) just leading to a very brief stop (initiated through GABA) and a subsequent reversal (due to electrical signaling via gap junctions).

This is a lengthy discussion, and we hope that the reviewer agrees that while it would give a more detailed explanation, we feel that what we wrote already in the paper (“We next assessed the role of neuropeptides in photoevoked RIS::ChR2 signaling by analyzing mutants lacking the Ca^{2+} -dependent activator protein for secretion (CAPS, encoded by *unc-31*), or the pro-protein convertase EGL-3. UNC-31 is required for secretion of (many of the) mature neuropeptides, while EGL-3 mediates processing of most if not all neuropeptide precursors⁵⁷. RIS photostimulation in *unc-31(e1304)* mutants still evoked stopping (Fig. 4A), thus release of neuropeptides mediating RIS effects may require factors other than UNC-31⁵⁸. However, *egl-3(gk238)* mutants were largely affected, and

stopped significantly less than WT:") basically contains the main argument why *unc-31* mutants retain RIS function, while *egl-3* mutants don't.

Reviewer #3 (Remarks to the Author):

The authors have done a thorough job of addressing my comments, and I remain enthusiastic about the paper and its findings, as detailed in my prior review. I have no further concerns or suggestions. It will be well-received and exciting for the field.

Michael Hendricks

Thank you!